# Conformational isomerism breaks the electrolyte solubility limit and stabilizes 4.9 V Ni-rich layered cathodes

Ziyang Lu[1,2,3], Huijun Yang [ORCID][2,3], Jianming Sun[1,2], Jun Okagaki [ORCID][2], Yoongkee Choe[2] & Eunjoo Yoo [ORCID][1,2] ✉

By simply increasing the concentration of electrolytes, both aqueous and non-aqueous batteries deliver technical superiority in various properties such as high-voltage operation, electrode stability and safety performance. However, the development of this strategy has encountered a bottleneck due to the limitation of the intrinsic solubility, and its comprehensive performance has reached its limit. Here we demonstrate that the conformational isomerism of the solvent would significantly affect the solubility of electrolytes. By transforming the configuration of solvent from cis-cis to cis-trans upon thermal triggering, we successfully break the solubility limit, and a beyond concentrated electrolyte with the lowest solvent-to-salt molar ratio of 0.70 is constructed. Transitions between cis-cis and cis-trans conformers are observed through Nuclear Magnetic Resonance (NMR) testing. The electrolyte consists entirely of anion-mediated solvation structures and promotes the formation of robust inorganic-dominated cathode electrolyte interphase. As a result, it enables stable cycling of 4.9 V-class $LiNi_{0.8}Co_{0.1}Mn_{0.1}O_2$ positive electrodes. Moreover, a high capacity of 151.2 mAh g$^{-1}$ can be maintained after 1000 cycles at cut-off voltage of 4.8 V. This work provides a chemical pathway to build new concept electrolytes working under harsh conditions.

One of the most important breakthroughs in battery technology is the development of concentrated electrolytes[1–3]. A typical representative is the development of "water-in-salt" or hydrate-melt aqueous electrolytes[4,5]. By simply increasing the concentration of the electrolyte, a high-voltage aqueous lithium (Li)-ion battery is successfully constructed. When this strategy is applied in non-aqueous electrolytes, the energy density is not only greatly improved but also exhibits some unexpected properties such as anti-corrosion[6–8], dendrite suppression[9–11], fire-extinguishing[12–14], extended operation window[4,5], and dramatically improved cycling stability[15,16]. From the prospective of constructing practical batteries with high energy densities, high voltage stability is among the top priorities for matching positive electrodes with high operating voltage and high capacity such as

typical Nickel (Ni)-rich layered positive electrodes[17–19]. Recently, researchers have made great progress in various electrolyte systems.

For ether electrolytes that are usually not resistant to high voltage, it can match 4.2 V or even 4.3 V Ni-rich positive electrodes and exhibit good cycling stability by increasing the concentration of electrolytes[20–23]. The improved high-voltage stability can be attributed to two aspects. Firstly, increasing the concentration lowers the HOMO (the highest occupied molecular orbital) level of the solvent, which reduces the reactivity under high voltage. In addition, the increased aggregation with concentration promotes the formation of inorganic-dominated cathode electrolyte interphases (CEIs) with high stability and quality due to the preferential decomposition of anions[24,25]. When using an ester electrolyte with better oxidative stability, the

[1]Graduate School of System and Information Engineering, University of Tsukuba, 1-1-1, Tennoudai, Tsukuba 305-8573, Japan. [2]Energy Technology Research Institute, National Institute of Advanced Industrial Science and Technology (AIST), 1-1-1, Umezono, Tsukuba 305-8568, Japan. [3]These authors contributed equally: Ziyang Lu, Huijun Yang. ✉e-mail: yu.eunjoo@aist.go.jp

concentrated electrolyte can further increase the anti-oxidation voltage to about 5 V. For example, the 5 V-class $LiNi_{0.5}Mn_{1.5}O_4$ (LNMO) positive electrode shows good cycling stability in a 7 M (mol $l^{-1}$) $LiN(SO_2F)_2$ (LiFSI)-fluoroethylene carbonate (FEC) electrolyte[26]. Using the same strategy, applying a LiFSI-dimethyl carbonate (DMC) electrolyte with solvent-to-salt molar ratio of 1.1:1, it achieves stable cycling for 100 cycles in 5 V-class batteries with LNMO positive electrodes[27]. In this case, the solvent-to-salt molar ratio is reduced to about 1, and no further research has been reported to break its limit either in aqueous or non-aqueous electrolytes (Fig. 1a). Even using such electrolyte, it is still far from being able to match more advanced positive electrodes and operate under more harsh conditions[28]. Compared with LNMO, the commercial $LiNi_{0.8}Co_{0.1}Mn_{0.1}O_2$ (NCM811) has higher energy density while more challenging due to the existence of $Co^{3+}$ and $Ni^{4+}$ with high catalytic activity, resulting in more severe electrolyte decomposition and phase transition under high operating voltage[29,30]. Raising the charging cut-off voltage to 4.8 V or even higher can basically fully extract the capacity, but the high voltage further deteriorates the interphase and structural stability[31,32]. Conventional methods such as introducing additives, interphase modification, etc. have little effect in promoting high voltage stability of NCM811[33,34], and less than half of capacities can be maintained over 100 cycles[35,36]. To cope with the instability under high voltage, the simplest yet most effective method is further increasing the concentration as reported and discussed above. Completely detaching HOMO orbitals from the solvent and

building a highly fluorinated interphase by using the highly concentrated electrolytes with low reactivity are expected to significantly improve the electrode stability under high voltages. However, limited by the intrinsic solubility of the specific salt in the selected solvent, breaking the solubility limit while maintaining its stability is extremely challenging both from an engineering and fundamental research point of view[37–39]. It requires exploring new concept electrolytes by introducing new chemical pathways and methods.

In this work, we demonstrate a chemical approach to push the solubility limit by manipulating the conformational isomerism of solvent (DMC) from *cis–cis* to *cis–trans* conformers. The configuration conversion can be achieved through thermal triggering, and the *cis–trans* configuration obtained under high temperature can be inherited even after returning to room temperature (Fig. 1b), which is completely different from the conventional supersaturated precipitation phenomenon. Conformational transitions between *cis–cis* and *cis–trans* are clearly revealed by $^1H$ nuclear magnetic resonance (NMR) of temperature gradients and Nuclear Overhauser Effect Spectroscopy (NOESY) NMR. And therefore, the molar ratio of solvent (DMC) to salt (LiFSI) is reduced to as low as 0.70, which is far lower than both aqueous and non-aqueous electrolytes and enables comprehensively improved electrochemical performance (Fig. 1c). It contributes to an entirely anion-mediated solvent structure with high oxidative stability and constructs a LiF-dominated CEI, which effectively inhibits the catalytic decomposition of electrolyte on the NCM811, dissolution of

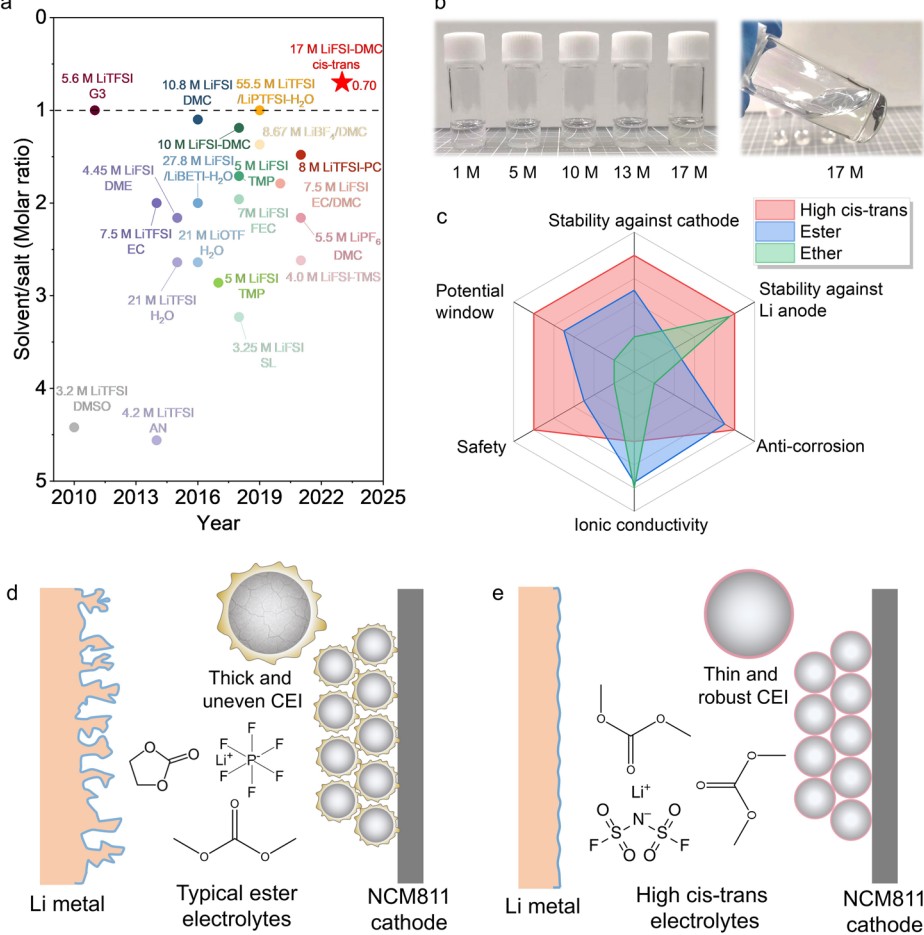

**Fig. 1 | Comparison of electrolyte properties. a** Comparison of recently reported works about highly concentrated electrolytes. **b** Images of various LiFSI-DMC electrolytes with different concentrations. **c** Radar chart of the advantages of concentrated electrolytes constructed by manipulating *cis–trans* configuration (compared with ester and ether electrolytes). **d**, **e** Failure mechanism of high voltage Li metal batteries using typical ester electrolytes (**d**) and high *cis–trans* electrolytes (**e**).

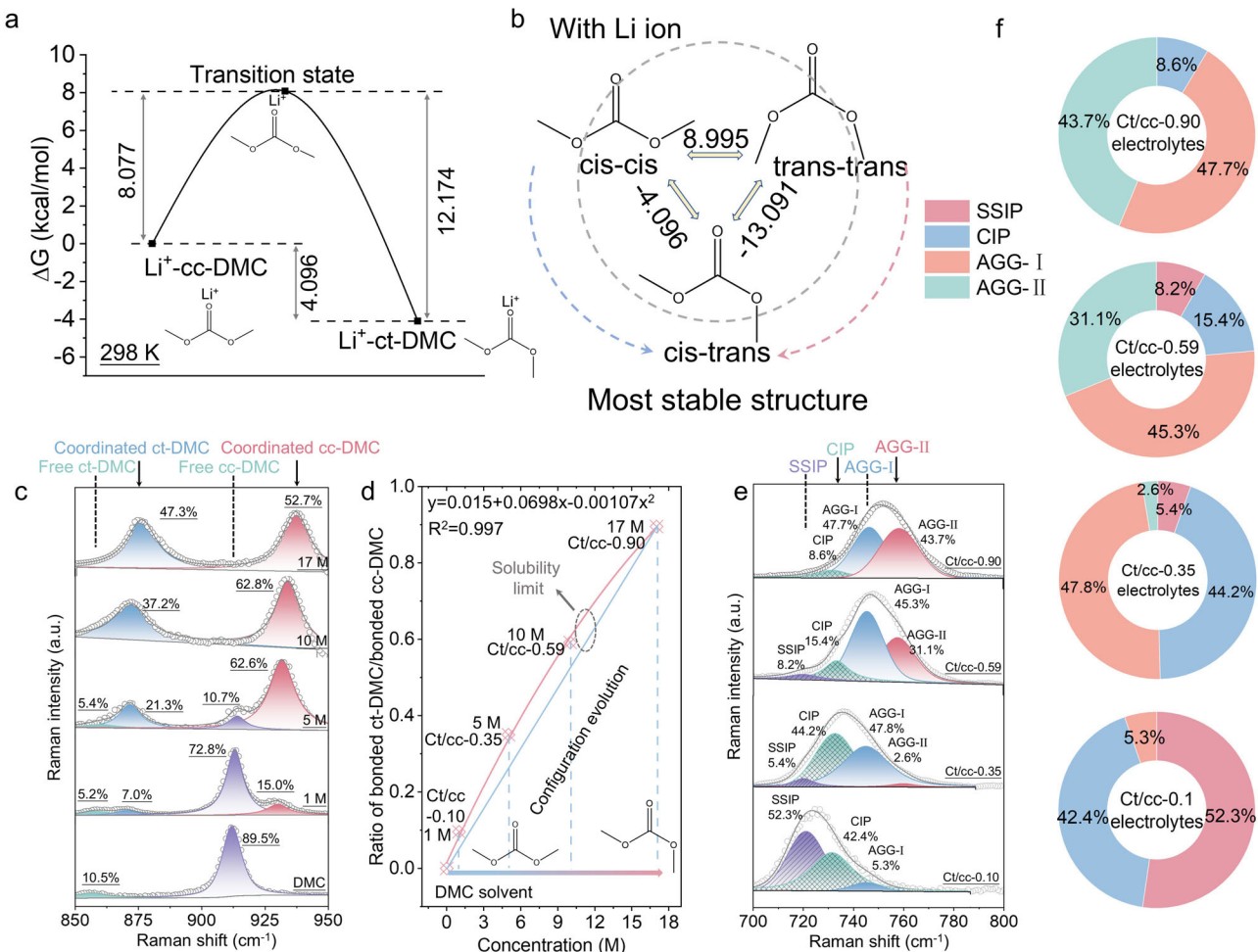

**Fig. 2 | Solvent structure analysis of concentrated electrolytes constructed by conformers. a** Transition state energies of different conformers of DMC solvent in the presence of Li⁺. **b** Schematic diagram of the interconversion of different conformers. **c** The fitting results of Raman spectra (O−CH₃ stretching of the DMC) of various electrolytes. **d** Relationship between the salt concentration (*x*-axis) and bonded ct-DMC/cc-DMC (*y*-axis). **e** The fitting results of Raman spectra (S−N stretching of the FSI⁻) of various electrolytes. The positions of the dashed lines and arrows in (**c**) and (**e**) correspond to peak positions of the relevant solvation structures. **f** Pie chart of solvation structure composition of different electrolytes.

transition metals and surface reconstruction (Fig. 1d, e). Therefore, even under high cut-off voltage of 4.9 V, a high capacity-retention of 77.9% can still be maintained after 100 cycles. At reduced cut-off voltage of 4.8 V, it delivers a high capacity of 151.2 mAh g⁻¹ after 1000 cycles with minimum capacity fading rate of 0.031% per cycle. The strategy of configuration isomerism regulation offers a pathway toward developing new concept electrolytes for aqueous and multivalent ion batteries operating under extreme conditions.

## Results

### Electrolyte properties

Typically, heating would increase the solubility of the solute in the solvent, but once the heating is stopped, the solute will inevitably precipitate again. Therefore, the molar ratio of solvent to solute can only be reduced to about 1 (Fig. 1a). Here we demonstrate that the configurational isomerism of solvent would significantly affect the solubility of electrolytes. The configuration of DMC gradually transforms into a *cis−trans* conformer from *cis−cis* upon thermal triggering. Therefore, a beyond concentrated 17 M LiFSI·DMC electrolyte is successfully constructed, which far surpass the previous 10 M LiFSI·DMC electrolytes[5,9]. Note that, the electrolyte concentration is calculated based on the volume of solvent rather than the total volume of the electrolyte. Interestingly, even after standing at room temperature for a week, the electrolyte remains clear and transparent without salt

precipitation (Fig. 1b). No nanoclusters were detected by dynamic light scattering (DLS), indicating that the 17 M electrolytes are homogeneous solutions rather than colloids or suspensions after adequate heating (Supplementary Fig. 1). But for the 17 M after heating for only 1 h, a peak at about 260 nm can be identified, suggesting that the LiFSI salt precipitates during the cooling and aging processes. The 17 M LiFSI·DMC electrolyte corresponds to a solvent/solute ratio of 0.70. To elucidate how conformational isomerism affects the solubility and stability of electrolytes, we performed density functional theory (DFT) calculations. For pure solvent, the typical *cis−cis* conformer has the most stable structure (Supplementary Fig. 2), which is consistent with our cognition. Additional energy is required to achieve conversion of the *cis−cis* state to other conformers. However, the *cis−trans* conformer becomes the most stable structure when taking the effect of Li⁺ into consideration for the real electrolytes (Fig. 2a). The Gibbs free energy of Li⁺-ct-DMC is 4.096 kcal/mol lower than that of Li⁺-cc-DMC. However, the transition from Li⁺-cc-DMC to Li⁺-ct-DMC requires crossing a high-energy transition state. It is well known that an edge-CH₃ in cc-DMC can be transformed into ct-DMC by rotating 180° along the adjacent C−O bond. The energy reaches its maximum value as the -CH₃ rotates about 90° along the C−O bond, which is the transition state shown in Fig. 2a. In this case, the kinetic energy barrier for the transformation of Li⁺-cc-DMC to Li⁺-ct-DMC is 8.077 kcal/mol, and the kinetic energy barrier for the transformation of Li⁺-ct-DMC to Li⁺-cc-

DMC is 12.174 kcal/mol. Actually, kinetic energy difference to cross the transition state leads to the difference in transformation kinetics between cc-DMC and ct-DMC and their coexistence at equilibrium. The thermodynamic barrier (4.096 kcal/mol) determines the ratio of Li$^+$-cc-DMC to Li$^+$-ct-DMC. The thermodynamic barrier increases from 4.096 to 4.194 kcal/mol as the temperature increases from 298 to 363 K (Supplementary Fig. 3), which is favorable to increase the proportion of ct-DMC. Therefore, increasing the temperature or increasing the heating time will change the ratio of Li$^+$-cc-DMC to Li$^+$-ct-DMC. The HOMO was also calculated for both cc-DMC and ct-DMC (Supplementary Fig. 4). The HOMO of the ct-DMC presents higher energy than the cc-DMC, which enables ct-DMC to be more easily coordinated with positively charged Li$^+$, thereby increasing the solubility of electrolytes. Therefore, we can understand why increasing the temperature changes the configuration of the solvent, and why the *cis–trans* state can still be retained after cooling to room temperature. First, the heating provides the necessary energy for pure solvents transform from *cis–cis* to *cis–trans* configurations. The *cis–trans* configuration is an asymmetric structure with larger polarity and higher dielectric constant than the *cis–cis* conformer. Therefore, increasing the temperature would improve the solubility by increases the proportion of *cis–trans* conformers. Even after the heating is stopped, the original *cis–trans* configuration obtained at high temperature can still be well preserved since it has the most stable structure with the lowest energy. Then, the solvation structure of this unique conformational isomerism-induced highly concentrated electrolyte was analyzed by Raman spectra. The coordinated *cis–cis* and *cis–trans* DMC, and free *cis–cis* and *cis–trans* DMC can be distinguished from the O–CH$_3$ stretching vibration band[40,41]. With the increase of concentration, the intensity of free *cis–cis* DMC located at 918 cm$^{-1}$ gradually decreases (Fig. 2c) after exceeding 1 M, while the bonded *cis–cis* DMC, *cis–trans* DMC configuration gradually increases. When the concentration reaches 17 M, the free *cis–cis* DMC completely disappears, and the bonded *cis–trans* DMC and *cis–cis* DMC dominant the solvation structure. The ratio of bonded *cis–trans* DMC/bonded *cis–cis* DMC can be obtained based on the O–CH$_3$ stretching fitting results, and the ratio increases with concentration as shown in the Fig. 2d. The ratio of bonded *cis–trans/cis–cis* DMC for 1 M and 10 M LiFSI-DMC electrolytes are 0.10 and 0.59, respectively, and the ratio reaches 0.9 when approaching 17 M. In order to present the relative proportion of *cis–trans* and *cis–cis* in the electrolyte, the 1 M, 10 M, and 17 M electrolytes are marked as ct/cc-0.10, ct/cc-0.59, and ct/cc-0.9, respectively. Correspondingly, peaks associated with FSI$^-$ can also see the evolution. As shown in Fig. 2e and Supplementary Fig. 5, there are distinct peaks between 720 and 760 cm$^{-1}$ for all electrolytes except the pure DMC solvent, which are generally attributed to S–N stretching mode of FSI[-27]. With the increase of concentration, the peak gradually shifts towards higher wave numbers, which originates from the ion association interaction transformation. It gradually evolves from solvent-separated-ion-pair (SSIP, free FSI$^-$) to contact-ion-pair (CIP, FSI$^-$ coordinate with one Li$^+$) and aggregate (AGG, FSI$^-$ coordinate with two or more Li$^+$) coordination. For ct/cc-0.10 electrolytes, SSIP and CIP dominate the solvation structure and coexist with a small amount of AGG-I (Fig. 2f). When the ratio of bonded *cis–trans* DMC/bonded *cis–cis* DMC is increased to 0.59 (10 M), the electrolyte contains 31.1% AGG-II, 45.3% AGG-I, 15.4% CIP, and 8.2% SSIP. As the ratio reaches 0.90 (17 M), the SSIP is completely eliminated, and AGG-I and AGG-II accounts for 47.7% and 43.7%, respectively. Interestingly, each LiFSI coordinates with 0.7 DMC solvent on average, which is an unprecedented coordination form. In this case, all solvation structures are anion-mediated, which enables some unexpected properties. For example, the high ct/cc-based concentrated electrolytes exhibits superior thermal stability and flame retardant ability compared with the commercial 1 M LiPF$_6$-EC/DEC electrolytes, which would contribute to a remarkably improved safety performance (see Supplementary Fig. 6). It is generally believed that

LiFSI is not conducive to the formation of protective passivation layer in dilute electrolytes, thereby corroding the aluminum (Al) current collector[42]. But in the high ct/cc-based concentrated electrolytes, the corrosion is effectively suppressed as observed in the photographs and optical microscope observation of glass fiber separators and Al current collectors (see Supplementary Figs. 7 and 8). Most importantly, it exhibits oxidative stability, which holds great potential in stabilizing high-voltage positive electrodes.

In order to identify the *cis–cis* and *cis–trans* conformers and analyze their impact on the evolution of the solvation structure, we conducted a systematic study by conducting Nuclear Overhauser Effect Spectroscopy (NOESY) NMR and $^1$H NMR as a function of temperature. At room temperature, two peaks can be observed (Fig. 3a), which correspond to $^1$H in D$_2$O and $^1$H in DMC. Note that, only a single peak appears for $^1$H in DMC since the six hydrogens in the *cis–cis* conformer have exactly the same chemical environment. Of course, insufficient resolution may result in the presence of conformers being buried. Interestingly, the peak splitting occurs as the temperature rises to 333 K. Obviously, the splitting of the peak is due to the formation of conformers different from *cis–cis*, namely *cis–trans*. When the temperature further increases, the splitting of this peak further intensifies. The intensity of the splitting peak on the left increases as the temperature increases, and its relative proportion also increases. Therefore, we attribute the left splitting peak to the *cis–trans* conformers, while the left splitting peak to *cis–cis* conformers. Then, the 2D NOESY spectrum was collected to analyze the interactions and exchange between *cis–cis* and *cis–trans* conformers. The $^1$H NOESY NMR spectra schematic diagram in Fig. 3b shows the information that can be interpreted. The peak at diagonal line is related to identical position of $^1$H, and the conversion between conformers can be reflected by the $^1$H at p1 and $^1$H at p2. The $^1$H NOESY spectra were collected at 333 and 363 K. As shown in Fig. 3c, there are 4 peaks, which correspond to the exchange of *cis–cis* and *cis–cis* conformers (cc-cc), *cis–trans* and *cis–trans* conformers (ct-ct), and the exchange between a pair of *cis–trans* and *cis–cis* conformers (ct-cc), respectively, indicating that *cis–cis* (4.270 ppm) and *cis–trans* (4.283 ppm) are transformed to each other. As the temperature further rises (363 K), four distinct peaks can still be observed, and peaks corresponding to the exchange of *cis–cis* (4.602 ppm) and *cis–trans* (4.617 ppm) become more significant (Fig. 3d), which is consistent with the 1D spectra in Fig. 3a. Finally, we compared the $^1$H NMR before and after the NOESY test (Fig. 3e, f). As expected, the peak intensity and relative proportion of the splitting peak (corresponding to *cis–trans*) on the left have increased whether at 333 K or 363 K since the NOESY test can be seen as a continuous heating process (lasting about 20 min), it promotes the transformation from *cis–cis* to *cis–trans*. These intuitive results illustrate the effect of temperature on conformer transitions. Then the $^1$H NMR spectra of pure solvent and 17 M electrolytes were collected after cooling to room temperature (Supplementary Fig. 9a). Only one single peak can be identified for the pure DMC solvent (without LiFSI salt) since H in cc-DMC has exactly the same chemical environment. However, two peaks can still be observed for the 17 M electrolytes because the ct-DMC is asymmetric with H in two different chemical environments, indicating that cc-DMC and ct-DMC coexist in the electrolytes. Therefore, we conclude that an important role of Li$^+$ is to maintain the stability of the ct-DMC configuration isomer. And then, $^1$H–$^1$H NOESY spectra were collected at room temperature (298 K). As shown in Supplementary Fig. 9b, four peaks can be observed, and peaks corresponding to the exchange of *cis–cis* (3.850 ppm) and *cis–trans* (3.862 ppm) were much weaker compared with the case tested at 333 K (Fig. 3d), indicating that *cis–trans* configuration is preserved in the presence of Li$^+$ after cooling, and exchanging between *cis–cis* and *cis–trans* was reduced with the decrease of temperature.

Combining spectroscopic analysis and theoretical calculations, the formation of highly concentrated electrolytes derived from

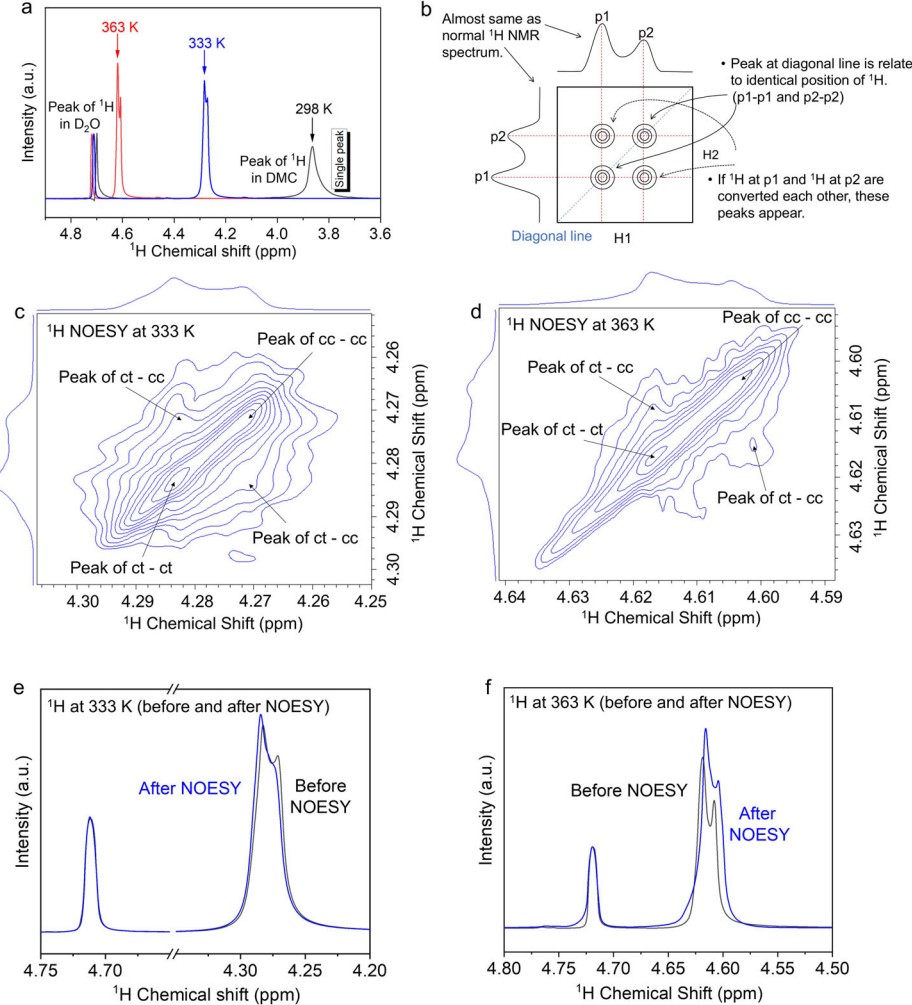

**Fig. 3 | Identifying the evolution of conformers by NMR tests. a** ¹H spectra of electrolytes at different temperatures. The intensity of ¹H in $D_2O$ is normalized. **b** Schematic diagram explaining the meaning of ¹H NOESY NMR spectra. **c** The fragment of ¹H–¹H NOESY NMR spectrum of electrolytes at 333 K in $D_2O$. **d** The fragment of ¹H–¹H NOESY NMR spectrum of electrolytes at 363 K in $D_2O$. **e** ¹H spectra of electrolytes before and after NOESY tests at 333 K. **f** ¹H spectra of electrolytes before and after NOESY tests at 363 K.

thermally induced conformers transitions can be clearly interpreted. The increase in temperature can promote the transformation of *cis–cis* to *cis–trans*, and the highly polar *cis–trans* conformer has higher solubility compared to the *cis–cis* configuration. In addition, the *cis–trans* conformer has lower energy than the equivalent *cis–cis* conformer after coordinating with $Li^+$. Therefore, the unique electrolyte configuration of the electrolyte obtained at high temperatures can be preserved even after cooling, which enables its application in Li-ion batteries at room temperature.

**High-voltage cycling stability in super-concentrated electrolytes**
The *cis–trans* conformational isomerism reconfigures the solvation configuration of the electrolyte, leading to a fully anion-mediated solvation structure. It detaches the HOMO orbital from the solvent, which is favorable to reduce the reactivity of electrolytes and build a highly fluorinated interphase to resist high voltage decomposition. To ensure the ionic conductivity of the electrolyte and wettability with the separator, the ct/cc-0.82 (15 M) electrolyte was used to evaluate various electrochemical performances in batteries. Supplementary Fig. 10a shows the oxidative stability of different electrolytes through linear sweep cyclic voltammetry (LSV) tests. With the increase of concentration, the initial oxidation decomposition potential gradually shifts towards higher voltage. For the ct/cc-0.82 electrolytes, slight electrolyte decomposition was observed only after the voltage

exceeded 6 V. Holding at 4.8 V for 24 h, the capacity accumulation due to oxidative decomposition is only one-tenth of that using 1 M $LiPF_6$-EC/DEC (see Supplementary Fig. 10b). The excellent high voltage stability of the high ct/cc-based electrolytes is further demonstrated by Potentiostatic Intermittent Titration Technique (PITT) floating tests (see Supplementary Fig. 11), which makes it possible to match high-voltage positive electrodes.

The nickel-rich layered material NCM811 is regarded as one of the most promising positive electrodes due to its high theoretical capacity (over 250 mAh g⁻¹), high redox potential and low cost[43]. As shown in Supplementary Fig. 12, the capacity can be effectively increased by extending the charging cut-off voltage. However, the high operating voltage would lead to severe interfacial side reactions, transition metal dissolution and phase transformation, which seriously affect the cycling stability of the battery[29,30]. At high cut-off voltage of 4.8 V, both the commercial 1 M $LiPF_6$-EC/DEC electrolytes and ct/cc-0.59 (10 M) electrolytes show severe capacity fading, voltage decay and fluctuating Coulombic efficiency (CE). Less than 150 mAh g⁻¹ capacity is retained after only 120 cycles at 0.5 C (1C = 200 mA g⁻¹) at about 25 °C (see Supplementary Fig. 13). However, for the ct/cc-0.82 electrolytes, even after cycling for 1000 cycles, a high capacity of 151.2 mAh g⁻¹ can still be maintained (Fig. 4a, b), corresponding to a capacity fading rate of 0.031% per cycle, which is lower than previous reports (Supplementary Table 1)[44–50]. Under more harsh condition of high temperature (60 °C),

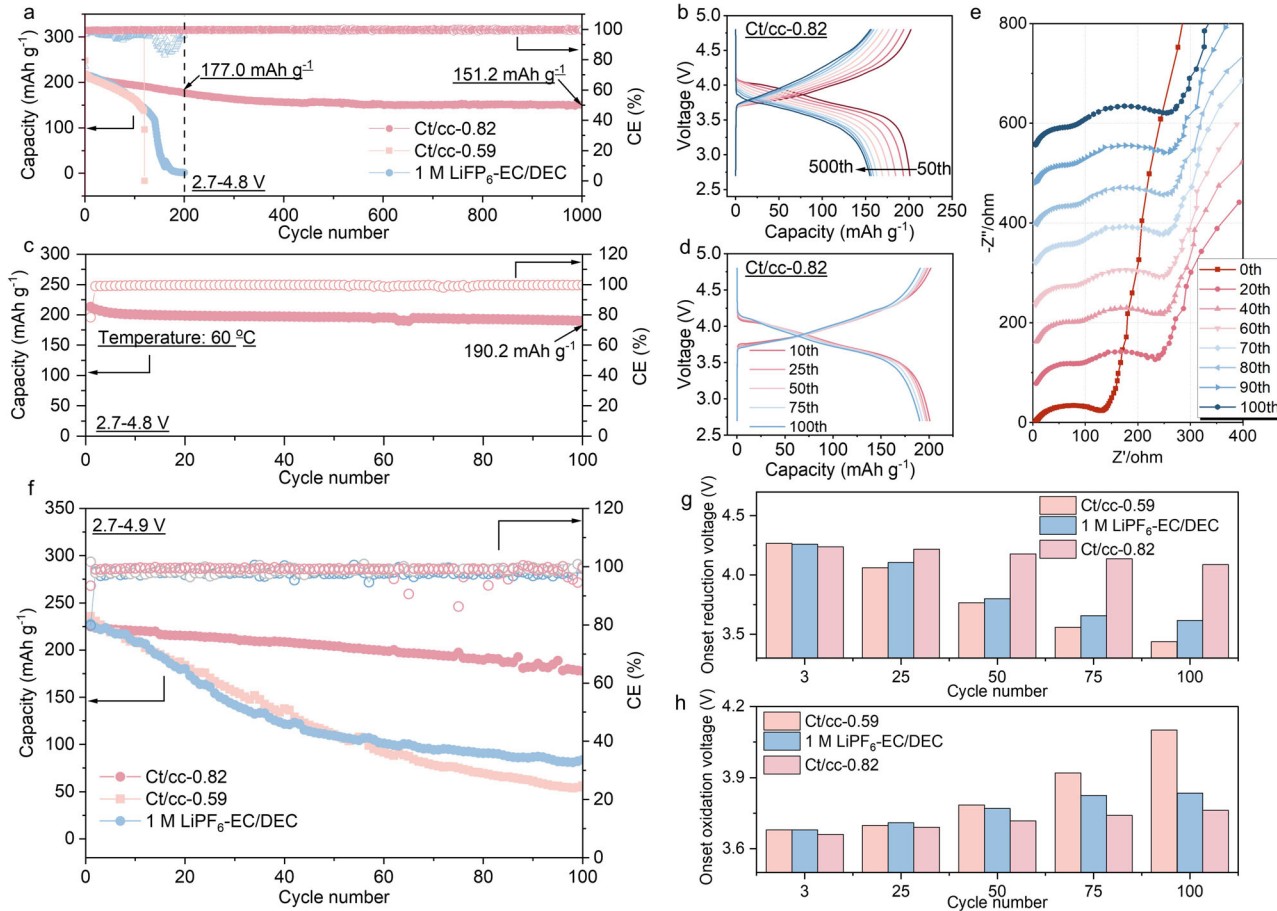

**Fig. 4 | Electrochemical performance of Li∥NCM811 cells using different electrolytes operating at high voltage. a** Cycling stability of Li∥NCM811 cells at cut-off voltage of 4.8 V with different electrolytes. The mass loading for NCM811 is 1.3 mg cm$^{-2}$. **b** Corresponding charge-discharge curves with ct/cc-0.82 electrolyte. **c** Cycling stability of Li∥NCM811 cells at cut-off voltage of 4.8 V and temperature of 60 °C. **d** Corresponding charge-discharge curves with ct/cc-0.82 electrolyte.

**e** Nyquist plots of high-voltage battery using ct/cc-0.82 electrolyte after various cycles. **f** Cycling stability of Li∥NCM811 cells at cut-off voltage of 4.9 V with different electrolytes. **g** The onset reduction voltage changes with the cycling number for Li∥NCM811 cells at cut-off voltage of 4.9 V. **h** The onset oxidation voltage changes with the cycling number for Li∥NCM811 cells at cut-off voltage of 4.9 V.

it also exhibits excellent cyclic stability and little voltage decay (Fig. 4c, d). At current density of 0.5C, a high capacity of 190.2 mAh g$^{-1}$ is maintained over 100 cycles. The improved cycling stability can be partially interpreted by the robust interphase generated by the super-concentrated electrolytes. As shown in Fig. 4e, both the resistance of Li$^+$ transport through CEIs and charge transfer resistance were basically unchanged over 100 cycles. For the commercial electrolytes, the charge transfer resistance is not only much larger than that in the case of ct/cc-0.82 electrolytes but also increases sharply within 5 cycles (see Supplementary Fig. 14). When the cut-off voltage further improved to 4.9 V, the capacity can be extracted almost completely. However, the cycling stability also further deteriorated. For batteries using 1 M LiPF$_6$-EC/DEC electrolytes and ct/cc-0.59 electrolytes, the capacity drops to 83.8 and 56.3 mAh g$^{-1}$, respectively, after 100 cycles at about 25 °C (Fig. 4f). By contrast, for the case using ct/cc-0.82 electrolytes, a high capacity of 177.8 mAh g$^{-1}$ is preserved after 100 cycles, corresponding to 77.9% capacity retention (see Supplementary Fig. 15). No H3b phase was observed from dQ/dV curves at such a high cut-off voltage (see Supplementary Fig. 16). In addition, neither the initial reduction voltage nor the initial oxidation voltage changed substantially during cycling for the case using ct/cc-0.82 electrolytes (Fig. 4g, h), indicating a good structural stability for the NCM811 positive electrode. However, the battery using 1 M LiPF$_6$-EC/DEC electrolytes shows significantly increased polarization during cycling. The initial reduction voltage negatively shifted by over 0.65 V after 100 cycles, suggesting that the

structure of NCM811 was severely damaged under such high operating voltage and could not maintain a stable voltage output.

To elucidate the stability of ct/cc-0.82 electrolytes under high operating voltage, the decomposition mode of electrolytes was systematically analyzed by applying operando Raman spectra. The battery configuration for the operando Raman test was shown in (see Supplementary Fig. 17). For the 1 M LiPF$_6$-EC/DEC electrolytes, the O–C–O ring bending mode of EC, symmetric P–F stretching of PF$_6^-$ (A1g), EC ring deformation and O–C–O ring bending of DEC can be identified (Fig. 5a)[51–53]. During charging, all peaks shift towards higher wave numbers with the increase of voltages, suggesting that the orientation and configuration of the electrolyte on the NCM811 electrode is changed when applying the electric field. For the peak of O–C–O ring bending of EC, as the voltage rises from OCP to 4.9 V, it moves from 718 to 722 cm$^{-1}$, and the peak of A1g vibration of PF$_6^-$ shifts from 740 to 745 cm$^{-1}$ (Fig. 5b), all of which are accompanied by a significant increase in peak intensity, indicating that both LiPF$_6$ salt and EC solvent involve in the decomposition. The same situation can also be observed in EC ring deformation and O–C–O bending. But EC ring deformation shows a larger wavenumber shift, suggesting that EC-associated solvent structures are more easily affected by the electric field leading to its decomposition preferentially over DEC. It is consistent with the previously reported theoretical and experimental results[54]. The same trend was verified in another test mode. Maintaining at 4.9 V for 30 min, all peaks shift to high wave numbers

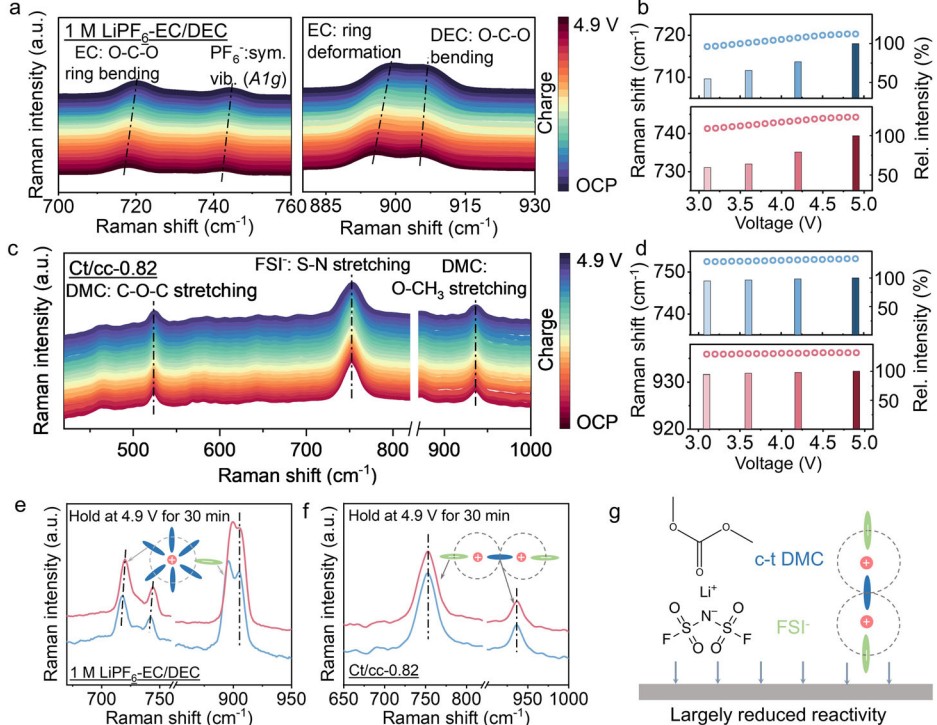

**Fig. 5 | Operando Raman measurements of electrolyte decomposition.**
**a** Operando Raman spectra collected during the first charge using 1 M LiPF$_6$-EC/DEC electrolytes. **b** Raman shift and intensity changes during charging. **c** Operando Raman spectra collected during the first charge using ct/cc-0.82 electrolytes. **d** Raman shift and intensity changes during charging. **e** Raman of 1 M LiPF$_6$-EC/DEC electrolytes before and after maintaining at 4.9 V for 30 min. **f** Raman of ct/cc-0.82 electrolytes before and after maintaining 4.9 V for 30 min. **g** Schematic diagram of the configuration of high-concentration electrolyte on the electrode. The operando Raman was conducted at constant charging current density of 0.5 C (100 mA g$^{-1}$).

obviously except DEC (Fig. 5e), which would result in the formation of thick and organic dominated CEIs, whereas a completely different situation was observed for the ct/cc-0.82 electrolytes. Typical Raman signals of solvents (O–C–O stretching and O–CH$_3$ stretching) and salts (S–N stretching) can be observed (Fig. 5c)[15,27]. These characteristic peaks just slightly shift with the increase of voltage. As shown in Fig. 5d, the peak shift of S–N stretching and O–CH$_3$ stretching is less than 1 cm$^{-1}$, and the intensity change is less than 6%. Similarly, when holding at 4.9 V for 30 min, the position and intensity of the correlation peak are basically unchanged (Fig. 5f), indicating that the configuration of various solvent structures in the ct/cc-0.82 electrolytes still maintain stable even under voltage as high as 4.9 V. It is not easy to induce oxidative decomposition (Fig. 5g). Different electrolyte decomposition modes lead to different CEI with different quality.

## CEI analysis and structure evolution of NCM811 under high operating voltage

Next, the transmission electron microscope (TEM) and X-ray photo-electron spectroscopy (XPS) were used to analyze the compositional and structural characteristics of CEIs formed in different electrolytes. For the NCM811 positive electrode cycled in typical 1 M LiPF$_6$-EC/DEC electrolytes (after 68 cycles), the particles were seriously pulverized (see Supplementary Fig. 18). A thick CEI layer over 31 nm was unevenly coated on the surface of the NCM811 (see Supplementary Figs. 19 and 20). High-resolution TEM and corresponding fast Fourier transform (FFT) images indicate that the CEI is mainly in an amorphous state, which is consistent with previous reports. As a striking contrast, the NCM811 particles (after 100 cycles) cycled in ct/cc-0.82 electrolytes remain intact (see Supplementary Figs. 21 and 22). The NCM811 particle is uniformly covered with a thin CEI layer of about 5 nm, and there are obvious inorganic crystalline regions, corresponding to the lattice spacing of LiF. Subsequently, the composition difference in CEIs was

further analyzed using XPS with depth etching. C, O, and F are the main components in CEIs. For the NCM811 positive electrode after 68 cycles in 1 M LiPF$_6$-EC/DEC electrolyte group, organic species dominate the CEI composition. On the outermost layer of NCM811, the organic component of O=C–O accounts for 7.0% (see Supplementary Fig. 23a). C-Metal bond is also observed, which can be attributed to the formation of metal carbides and related C-Metal compounds during electrolyte decomposition on the NCM811. After 12 min of etching, the proportion of organic component of O=C–O increased to 17.9%, and the content of C-Metal can also see an increase. It stems from the increased reactivity as it approaches the NCM811 surface, which leads to more severe electrolyte decomposition and induces associated side reactions. The C-Metal bond is always accompanied by the phase transition as observed in TEM images[55]. A similar pattern can be observed for O 1$s$ (see Supplementary Fig. 23b). The components of C=O and C–O are always dominant, and only a small amount of Li$_2$O coexists. The difference is that the content of inorganic Li$_2$O components increases with etching, and this phenomenon also exists in F 1$s$. The proportion of LiF increased to 54.6% after etching for 12 min from pristine 18% (see Supplementary Fig. 23c). Therefore, we can deduce that the composition of CEIs is dominated by organic species, and the distribution of various components is uneven. In contrast, for the NCM811 cycled in ct/cc-0.82 electrolytes, no C=O–C peak was observed after 100 cycles. The content of O=C–O was also much lower than the CEI formed in conventional commercial electrolytes (see Supplementary Fig. 23d). In addition, the C-Metal bond is absent for the case using ct/cc-0.82 electrolytes, suggesting that side reactions related to electrolyte decomposition are effectively suppressed. For the O 1$s$ spectrum, Li$_2$O is always dominant, and its content increases to 78.1% after 12 min of etching (See Supplementary Fig. 23e). No organic C-Fx bonds were detected from the F 1$s$ spectra. Only a distinct LiF peak can be observed during the continuous etching process (see

Supplementary Fig. 23f). The atomic content of F in the CEI formed in the ct/cc-0.82 electrolyte is about 4%, which is much higher than that in conventional commercial electrolytes (see Supplementary Fig. 24). Based on the fitting results of XPS, we can conclude that the CEI formed in the ct/cc-0.82 electrolyte is more uniform, dominated by LiF and $Li_2O$, and coexisting with a small amount of organic components. The dominant inorganic component is mainly derived from the decomposition of salts. The difference in CEI properties can also be observed from in situ electrochemical impedance spectroscopy (EIS) spectra. When approaching the charge cut-off voltage, the charge transfer resistance rises sharply and reaches 7366 Ω for the battery using 1 M $LiPF_6$-EC/DEC electrolytes, which is much larger than that using ct/cc-0.82 electrolytes (2109 Ω) (see Supplementary Fig. 25), which is related to the difference in the quality of CEI formed in different electrolytes. Subsequently, the ex situ FT-IR tests further verified the compositional characteristics of the CEI formed in these two electrolytes. For the case using 1 M $LiPF_6$-EC/DEC, the intensity of organic-related peaks gradually enhance as the charging voltage increases (see Supplementary Fig. 23g). The peak further intensifies after discharging, indicating that the CEI formed under high voltage is not reduced at low voltage. The intensity of those peaks further increases in subsequent cycles, indicating that the formed CEI cannot effectively block the electrolytes from decomposition. By contrast, for the case using ct/cc-0.82 electrolytes, the increase in charging voltage basically does not trigger electrolyte decomposition (see Supplementary Fig. 23h). Even after 5 cycles, the curves almost overlap.

In addition to the differences in SEIs, the structural stability under high voltage is also quite different. The TEM clearly reveals the structure evolution of the NCM811 particles. For the case using typical 1 M $LiPF_6$-EC/DEC electrolytes, multiphase structure with a gradient can be observed (see Supplementary Fig. 26a), which corresponds to the formation of thick CEIs and phase transition. The rock salt phase and the layered oxide structure can be well distinguished by both TEM and corresponding FFT images (see Supplementary Fig. 26b–e), indicating that the instability of the interface under high voltage leads to severe structural damage, which results in rapid capacity decay. By contrast, the NCM811 positive electrode cycled in ct/cc-0.82 electrolytes shows a homogeneous single-phase structure (see Supplementary Fig. 26f). The typical layered structure is well preserved, and no rock-salt phase can be detected (see Supplementary Fig. 26g–j), suggesting that the surface reconstruction has been effectively suppressed. To reveal the phase transition upon charge-discharge under high cut-off voltage of 4.9 V, ex situ XRD measurements were conducted on the 11th cycle for 1 M $LiPF_6$-EC/DEC and ct/cc-0.82 electrolytes. The typical (003) peak exhibits a regular shift in both electrolytes (see Supplementary Fig. 27a, b), which is positively correlated with the evolution of $c$ lattice parameters. During charging, the (003) peak shifts first to high angles and then to low angles (see Supplementary Fig. 27a), which corresponds to typical phase transitions of H1 → M → H2 → H3a for the case using ct/cc-0.82 electrolytes[56]. But a new phase of H3b appears for the positive electrode using 1 M $LiPF_6$-EC/DEC electrolyte (see Supplementary Fig. 27b). The (003) peak has a larger shift towards low angles than that in the ct/cc-0.82 electrolytes during charging, demonstrating a larger $c$-lattice ($\Delta c$) parameter change. In addition, the H3b phase always exists during the charge (see Supplementary Fig. 27c) and discharge (see Supplementary Fig. 27d) process. The H3b phase was formed under continuous accumulation during cycling as it was not observed in the initial cycle, which would eventually lead to structural degradation and capacity fade. In addition, the volume change during the charging process is also calculated. For the case using 1 M $LiPF_6$-EC/DEC electrolyte, the expansion in the $c$-lattice direction is 2.1%, which is significantly larger than that using ct/cc-0.82 electrolytes (1.2%) (see Supplementary Fig. 27e). A similar trend can also be observed from a-lattice parameter variation (see Supplementary Fig. 27f). The maximum changes of the a are 2.2% and 1.5%, respectively in 1 M $LiPF_6$-EC/DEC electrolytes and ct/cc-0.82 electrolytes. In addition, the high voltage induces severe dissolution of transition metals, which easily shuttles to the Li metal negative electrode and undermines its stability. The signal of Mn 2$p$ can always be detected on the cycled Li metal negative electrode in 1 M $LiPF_6$-EC/DEC electrolytes (see Supplementary Fig. 28). And under continuous etching, the signal persists. However, a negligible peak can be observed for the Li metal negative electrode cycled in ct/cc-0.82 electrolytes, indicating that the dissolution of transition metals was effectively suppressed. Those results indicate that the stable interphase constructed by the high cis/trans-based electrolyte effectively stabilizes the structure of NCM811 under high voltage.

In conclusion, manipulating the configurational isomerism of the solvent is proposed to break the limit of solubility and construct beyond concentrated electrolytes. By transforming the configuration of solvents from $cis−cis$ to $cis−trans$, a 17 M electrolyte is successfully constructed with the lowest solvent-to-salt molar ratio of 0.70, which enables a fully anion-mediated solvation structure with dramatically enhanced oxidative stability. The fully anion-mediated electrolyte generates a robust CEI dominated by LiF, which effectively suppresses the catalytic decomposition of electrolytes on the surface of NCM811 and transition metal dissolution under high voltage. In addition, the surface reconstruction and phase transition to unfavorable H3b are eliminated by the integrated interphase engineering and electrolyte regulation. Applying the proposed electrolyte in the high voltage Li‖NCM811 battery, it can stable operate for 1000 cycles with minimum capacity decay rate at high cut-off voltages of 4.8 V, and a high capacity-retention of 77.9% can still be preserved after 100 cycles as the cut-off voltage increases to 4.9 V. Furthermore, a high capacity of 190.2 mAh g$^{-1}$ is maintained over 100 cycles under harsh condition of high temperature (60 °C). The electrolyte constructed by conformational isomerization comprehensively improves the high voltage and high-temperature stability of Li metal batteries, which provides a pathway for developing more practical electrolytes working under harsh conditions.

## Methods
### Electrolytes
The electrolytes with different concentrations were prepared by dissolving Lithium Bis(fluorosulfonyl)imide (LiFSI) (Wako Pure Chemical Industries, Ltd, Guaranteed Reagent) in Dimethyl carbonate (DMC). The LiFSI was dried by heating under vacuum at 80 °C to remove any moisture that may be present in an oven overnight. The DMC was dried using activated 4A zeolite molecular sieves (Wako Pure Chemical Industries Ltd) before use. For highly concentrated electrolytes over 10 M, it was prepared by stirring with heating at 90 °C for more than 24 h. The commercial carbonate electrolyte of 1.0 M lithium hexafluorophosphate ($LiPF_6$) in ethylene carbonate and dimethyl carbonate (EC/DEC, 1:1, wt/wt), was purchased from Tokyo Chemical Industry.

### Electrodes preparation
Single crystal $LiNi_{0.8}Co_{0.1}Mn_{0.1}O_2$ (NCM811) was used in this work. The positive electrode was prepared by mixing NCM811, powder super P conductive carbon (Lion Specialty Chemicals), and PVDF (Du Pont-Mitsui)-N-methylpyrrolidine (NMP, Sigma-Aldrich, 99%) binder solution (PVDF/NMP = 1:32) in a ratio of 8:1:1 and stirring overnight. Then, the obtained sticky slurry was coated on an Al foil using a scraper and vacuum dried at 100 °C for 24 h. For coin-cell measurements, the electrodes were punched into a plate with diameter of 11 mm and the mass loading controlled at 1–2 mg cm$^{-2}$.

### Cell assembly and electrochemical measurements
All batteries were assembled in a glove box with dew point of below −90 °C and $O_2$ content below 7 ppm. 2032 coin cells (Hohsen) were used to evaluate the high voltage performance of NCM811 in different

electrolytes. The battery is assembled by sequentially stacking Li foil (about 0.4 mm), glass fiber filter (GF/A, Whatman), and NCM811 positive electrode. The diameters of the GF/A separator, the NCM811 positive electrode and the Li metal negative electrode are 16 mm, 11 mm, and 12 mm, respectively, and the amount of electrolyte was controlled at 50 μL. The same size is used for in situ Raman assembly, except that a small hole with a diameter of 4 mm is punched on the center of Li foil and the GF/A separator to collect the Raman signal on the positive electrode. The cells were tested between 2.7 and 4.9 V. Li‖Cu half batteries and Li‖Li symmetrical batteries are assembled to study the stability of Li metal negative electrodes under 60 °C. For the coin cell tests, the galvanostatic electrochemical measurements were carried out using the battery test system HJ1001SD8 (Hokuto Denko) at 25 °C and 60 °C. All the cells were maintained at OCP for 8–10 h before electrochemical tests. The potentiostatic intermittent titration technique (PITT) floating test, electrochemical impedance spectra (EIS) and linear sweep cyclic voltammetry (LSV) tests were carried out using (Potentiostat/Galvanostat PGSTAT30, Autolab Co. Ltd., Netherlands) work station at 20–25 °C. EIS was recorded in a frequency range from 0.01 Hz to $10^6$ HZ. The operando Raman was conducted in a customized in situ Raman cell (Hohsen Corp., Osaka, Japan) at charging current density of 0.5 C ($100 \, mA \, g^{-1}$). A thin quartz window with thickness of 0.5 mm was fixed on the top of the in situ Raman cell. Raman signal was collected through the quartz window.

## Characterization

The morphology analysis was conducted by scanning electron microscopy (SEM, JEOL JSM-6380LV FE-SEM). Elemental analysis of the cycled NCM-811 positive electrodes and Li metal negative electrodes was conducted on X-ray photoelectron spectroscopy (XPS) using a VG scientific ESCALAB 250 spectrometer with monochromic Al Kα excitation (1486.6 eV). X-ray diffraction (XRD) measurements were performed on a Bruker D8 Advanced diffractometer fitted with Cu Kα ($\lambda = 1.5406$ Å) radiation. All electrodes extracted from cycled cells were washed three times with dimethoxyethane (DME, Sigma-Aldrich, 99%) in glove box filled with Ar gas to remove residual electrolytes and then dried by evaporation in a vacuum chamber. Then, the electrodes were moved back to the glove box and sealed in a glass box filed with Ar gas. During transfer, the samples were always protected by Ar. Transmission electron microscopy (TEM) was conducted on a JEM-2100F (JEOL, Japan) to analyze the CEI and structure change of NCM811. The Raman spectra of electrolytes and NCM811 positive electrodes were carried out on a JASCO microscope spectrometer (NRS-1000DT). An air-cooled He-Ne laser at 632.8 nm wavelength was focused on the sample through a 50 × long working distance lens (Olympus America Inc.). The acquisition time was 90 s with 2 accumulations during Raman spectrum collection. Two places are chosen to ensure the accuracy and repeatability of Raman spectra. After the peak positions of each solvation structure are determined, the relative proportions of various solvation structures can be obtained through fitting. Fourier-transform infrared (FTIR) measurements were carried out on a FT/IR-6200 spectrometer (JASCO Corp.). Nuclear Magnetic Resonance (NMR) spectra were recorded using a Bruker Avance III 500 MHz and Bruker 600 MHz "ASCEND AVANCE III HD". The $^1H$ NMR spectra of samples in different temperature (298, 333, and 363 K) were recorded in $D_2O$ (99.9 atom% D, Wako Chemicals). For the $^1H$ NOESY measurement, the inner tube was filled with $D_2O$ and the spectrum was collected at 333, 363, and 298 K. Each test is held at aimed temperature for 5 min, and each test lasts ~20 min. The Dynamic Light Scattering was conducted on an OMEC NS-90Z Nanoparticle Size and Zeta Potential Analyzer using malvern curvettes (DTS0012).

## Computational details

All electronic structure calculations were conducted within the framework of density functional theory (DFT). Geometry optimizations were performed using the wB97X-D3[57]. As for basis sets, we employed def2-SVP sets[58]. Transition state search was performed using the Nudged Elastic Band (NEB) method[59]. Additional frequency calculations were carried out to confirm the nature of optimized structures. All the calculations utilized ORCA software (version 5.0.1)[60]. The Gibbs free energies can be calculated by the following equation:

$$G(T) = H(T) - TS(T)$$
$$= \varepsilon_{ele} + ZPE + G_{0 \to T}$$

Where G is Gibbs free energy, H is enthalpy, T is temperature, S is entropy, $\varepsilon_{ele}$ is electronic energy, ZPE is acronym of zero-point energy. The analysis was conducted using Shermo software[61].

## Data availability

All data supporting this study and its findings are available within the article and Supplementary Information. Additional supporting data of this study are available from the corresponding author on request. Source data are provided with this paper.

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

## Acknowledgements
We thank G. Wu (University of Tsukuba, Japan) for his help in the Raman and SEM characterizations and general discussion.

## Author contributions
Z. Lu and E. Yoo designed the research. Z. Lu performed the electrochemical testing and most characterization. Z. Lu and H. Yang contributed new reagents/analytic tolls. Z. Lu and E. Yoo analyzed data. J. Sun helped with ex situ XRD testing and related analysis. J. Okagaki conducted the NMR tests. Y.K. Choe conducted the DFT calculation. Z. Lu and E. Yoo wrote the paper; all authors discussed the results.

## Competing interests
The authors declare no competing interests.
