## [Transparent Peer Review file · Nature Communications]

Conformational isomerism breaks the electrolyte solubility limit and stabilizes 4.9 V Ni-rich layered cathodes

Corresponding Author: Professor Eunjoo Yoo

This manuscript has been previously reviewed at another journal. This document only contains reviewer comments, rebuttal and decision letters for versions considered at Nature Communications.

Version 0:

Reviewer comments:

Reviewer #1

(Remarks to the Author)

The authors have addressed our concern (Nature Chemistry submission) by quantitatively measuring the presence of different conformational isomers of DMC. However, they have not confirmed the thesis for which they irreversibly change the isomerism from cis-cis to cis-trans at room temperature in presence of lithium ions. The DFT calculations of Figure 2a suggest that, in presence of lithium ions Li^+ , the cis-trans isomer is a lower minimum energy configuration than cis-cis, and therefore, being the energy barrier relatively low (4 kcal/mol \sim 16 kJ/mol) it can be achieved by thermal triggering. The NOESY NMR is able to confirm this. The data presented show that DMC, upon thermal triggering, can swap between cis-cis and cis-trans conformations, however they have not shown that the 17M solution in DMC presents univocally cis-trans isomers. We understand that the salt is not soluble at 17M in DMC at room temperature, but the authors claim that it is at higher temperature, and that the solution stays stable at room temperature because of the transition between cis-cis and cis-trans. We therefore suggest to perform the NOESY NMR of the 17M solution at high temperature, and then do the measurement, rigorously on the same sample, at RT again to show that the cis-trans is still the prevalent isomer.

The characterization reported here is still insufficient to prove their claims.

Response to the Referees 1 – “NOESY-NMR clearly identifies two isomers and temperature-driven interconversion.”
The two isomers are interchangeable with the increase of the temperature, and this was expected. The data provided do not show that in presence of lithium, one isomer prevails over the other, nor that is the reason for the increased solubility.

Page 7 line 13 – “The ^1H NOESY spectra were collected at 333K and 363K.”
In order to prove irreversible isomer transformation, these measurements should include the RT after the thermal triggering.

Page 7 line 7 – “Interestingly, even after standing at room temperature for a week, the electrolyte remains clear and transparent without salt precipitation (Fig. 1c).”

Note there are two Fig. 1c.

By comparison with the 1M – 13 M solutions, the 17 M looks opaquer. Due to the high concentration of salt, this is to be expected, however we advise DLS or SAXS/WAXS measurements to exclude the presence of micro clustering and cation-anion recombination, that the authors confirm will occur as stated by the response to comment 4.

Page 8 line 8 – “With the increase of concentration, the intensity of free cis-cis DMC located at 918 cm^{-1} gradually decreases (Fig. 2d) after exceeding 1 M, while the bonded cis-cis DMC, cis-trans DMC configuration gradually increases.”
The Raman data and the authors confirm that there is a significant portion of cc-DMC that is interacting with the salt namely ‘bound cc-DMC’. This excludes than the isomeric transition to be the only reason for the increased solubility.

Page 9 line 6 – “In this case, all solvation structures are anion-mediated, and the solvation structures are cross-linked to each other via Li^+ or FSI - to form a polymer-like three-dimensional structure.”

Response to Referee 9 – The authors in the response to the referee stated: “The polymer-like three-dimensional structure we describe here is not a real polymer structure.”

We agree with the authors and we believe the text needs to be rephrased because the characterisation provided is insufficient to confirm or exclude the solvation structure proposed.

Response to Referee 9 – The structure we describe here is a three-dimensional interconnected structure formed by the tight combination of FSI⁻ and Li⁺.

Without further analysis the proposed structure cannot be confirmed as, at such concentrations, the solution can easily be microscopically inhomogeneous. We believe that the analysis does not support the result and that there is no evidence of the proposed solvation structure.

Page 12 line 3 – “Combining spectroscopic analysis and theoretical calculations, the formation of highly concentrated electrolytes derived from thermally induced conformers transitions can be clearly interpreted. The increase in temperature can promote the transformation of cis-cis to cis-trans, and the highly polar cis-trans conformer has higher solubility compared to the cis-cis configuration.”

The statement that the cis-trans has higher solubility than cis-cis is still unsupported. This needs a reference or solubility measurements. The change in isomorphism could be one, but not the only, reason of the increased solubility.

Page 12 line 7 – “In addition, the cis-trans conformer has lower energy than the equivalent cis-cis conformer after coordinating with Li⁺. Therefore, the unique electrolyte configuration of the electrolyte obtained at high temperatures can be preserved even after cooling, which enables its application in Li-ion batteries at room temperature.”

The spectroscopic measurements do not support this point as they do not show that in the presence of Li⁺ the cis-trans configuration is preserved after cooling. The NOESY shows an interchange between cis-trans as a function of the temperature, as expected.

Page 10 Figure 2e – We believe that the fit to the Raman data should be presented here. It is not clear what the dotted lines are indicating (spectra regions?) as they do not correspond to any peak position. This point can be addressed by expanding the caption. The ratio between ct and cc needs to be quantitatively evaluated. In the highest regime ct/cc is still < 1, which means that there is still a greater ratio of cc than ct.

Version 1:

Reviewer comments:

Reviewer #1

(Remarks to the Author)

The authors have now addressed the comments satisfactorily. I recommend the article for publication. I have a few minor comments that could help further improve the manuscript:

- ‘As shown in Fig. 4g, the sharp peak (1.85 Å) of Li⁺-O in DMC corresponding to the primary solvating shell, and the Li⁺-O in ct-DMC has higher intensity than the cc-DMC, demonstrating that the Li⁺ is more easily coordinated with ct-DMC than with cc-DMC.’
- We note that increased intensity does not always mean higher coordination as the coordination is the integral of the peaks. We would advice to say ‘suggesting’ rather than ‘demonstrating’.
- Radial not radiation distribution function
- We believe that the response to Comment 4 could be rephrased more positively, and we do not think that the DFT calculations and the Raman data are contradicting each other. The transition from cc to ct is confirmed by the NMR and both isomers exist in solution. We believe that it is one reason, but not the only reason for increased solubility. Another reason as the authors mention is the reduction of steric hindrance. We therefore suggest not to mention intermediates that are not consistent with the data showed.

Reviewer #3

(Remarks to the Author)

This work proposed a conformational isomerism mechanism to break the electrolyte solubility limit and stabilize 4.9 V Ni-rich layered cathodes. The solvation structures were comprehensively probed by both experimental and theoretical characterizations. Further, excellent electrochemical performances were obtained due to the super-high salt concentration. I go through both the revised manuscript and rebuttal materials and personally think this work can hardly meet the high standards of Nature Communications. Details reasons are provided as follows.

One critical point is that whether the cis-trans solvent structures exist at room temperature. A lot of experimental characterizations including NMR were conducted to probe this point. However, there are several major issues for the simulation results.

- (1) Figure 2a is used to explain the energy difference of ct and cc structures. However, the influence of entropy and temperature is not included in the total energy. If ct structure is always more stable than cc structure when coordinating with a Li ion, the calculation results can not explain the different solvation structures induced by different heating times.
- (2) The MD results are not reliable. The box is too small to get enough sampling, which can also be verified by the RDF plots. The $g(r)$ should be converged to 1 when r is large enough. A mistake may be made while analyzing the MD results. Besides, how are CC- and ct-DMC structures controlled in simulations? The MD results should deliver equilibrium sampling structures. The sharp peak of Li-O profiles is around 1.85 Å, which is even smaller than that in Li₂O crystals. Therefore, wrong or unsuitable simulation parameters may be adopted.

(3) If DMC structures are not fixed during MD structures, it is suggested to analyze their percentages under different temperatures and salt concentrations.

Therefore, the current simulation results can hardly support the conclusions of solvation structure analyses in this work. Besides, there are some other issues that should be further considered.

(1) Figure 1a compares the solvent-to-salt ratio of various electrolytes. However, it should be noted that the calculation method of salt concentration can be different in different papers. Especially, the mix volume effect can have a significant influence on the calculated salt concentration.

(2) "A solvent-to-salt ratio of 1 seems to be the solubility limit of electrolytes." A solvent-to-salt ratio of 1 is highlighted. However, why such a ratio should be the limitation of solubility? The underlying chemistry and materials science should be explained.

(3) It is suggested to analyze the HOMO of different cc- and ct-DMC structures to demonstrate their electronic differences.

Version 2:

Reviewer comments:

Reviewer #3

(Remarks to the Author)

Although the authors added a lot of calculations to revise the manuscript, previous issues have been well addressed. Therefore, I still do not think the current manuscript can meet the high standards of Nature Communications.

1. In previous Comments 1, "the thermodynamic barrier increases from 4.096 kcal/mol to 4.194 kcal/mol as the temperature increases from 298 K to 363 K". The thermostability energy difference between cc-DMC and ct-DMC is too small, which can not explain the obvious solubility salt differences among different temperatures.

2. The NPT and NVT ensembles should deliver the same result when the structures are sampled enough. According to comments 2, the unreasonable RDF results may indicate that the structures were not sampled sufficient or there is a mistake during calculation or data analysis, which is not acceptable for a high-quality Nature Communications paper.

3. The review did not highlight the difference between classical MD and AIMD. The key point is that the results should be reasonable. It would be appreciated that the simulation results agree with the experimental results or the simulation results can explain the experiments well.

4. In comment 3, the ratio of ct-DMC ranges from 3.5% to 3.8% from 298 K to 363 K. The ratio difference is very small, which can not explain the experimental results of conformational isomerism regulation.

Reviewer #4

(Remarks to the Author)

Summary:

This manuscript presents a novel approach to enhancing the performance of lithium metal batteries by manipulating the solvent's configurational isomerism to create a highly concentrated electrolyte. By switching from a cis-cis to a cis-trans configuration of the DMC solvent, the authors successfully developed a 17 M electrolyte with a low solvent-to-salt ratio, resulting in a fully anion-mediated solvation structure with improved oxidative stability. This electrolyte significantly enhances the stability and performance of high-voltage Li||NCM811 batteries, even under extreme conditions, such as higher operational temperatures and high cut-off voltages.

However, the description of the simulations are not up to the standards of Nature Communications, so I cannot recommend the publication of this manuscript in this journal.

Major Issues

The reliability of the NVT ensemble is not described. In Fig. 4, which shows a "snapshot", there is considerable vacuum in the space. The authors simply say that "pressure was maintained under 1 atm during the simulation." More precise language is needed here. I assume they mean that the pressure was maintained at 1 atm, but it can also be interpreted that it was below this pressure. The authors describe that there were volume fluctuations during the trajectories, but there is no description of how large these fluctuations were. How closely did the density of the electrolyte match experimental values?

How was the cc-ct ratio monitored in the AIMD simulation? The short simulation time may have resulted in only a minor switch from cc to ct molecules in the trajectory, which may need to be acknowledged.

Aggregation in the electrolyte is not visually distinguishable in Figure 4. (as mentioned in Fig 4a vs. 4d); the same is true for NVT (Fig. S10), which shows no agglomeration. It needs to be explained to be convincing. Additionally, could this have been a result of the NPT ensemble?

Even for an NPT-based simulation, RDF should approach 1. The authors' reason that this may not be the case for NPT is only valid if the distances are fixed in the starting image of the trajectory and, consequently, there is a significant shift in the volume during the equilibration process. For a given snapshot, it should go to 1. This may need to be acknowledged. Also, it should be mentioned if the RDF curves have been smoothed. The peaks are expected to be more spiky unless smoothed for a cell of this size.

Fig. S26 color/voltage legend needed?

The equilibration criteria and related data for the trajectories need to be specified. Specifically, please clarify the phrase 'actual equilibrium state' in the statement "It should be noted that our simulation results did not reach the actual equilibrium state due to the limitations of computing resources and time." provided in the rebuttal for Review 2. This work will only be acceptable if the system is equilibrated computationally (energy and temperature) to extract conclusions from the calculated trajectory.

Minor Issues

Abstract - Line 20 - "Transitions between cis-cis and cis-trans conformers were visually observed through Nuclear Magnetic Resonance (NMR) testing": "Visually" from NMR did not make sense.

Fig. 1 has two panels labeled c)

In the main text there is a description (on p. 8) saying the HOMO has a "much higher energy" in ct-DMC than in cc-DMC. I disagree that that 43 meV counts as a considerable difference. Also, regarding Supplementary Fig. 4, it shows side and top views of MOs, but it doesn't say which orbitals we are looking at (HOMO or LUMO, or which is which).

The panel order in Fig. 3 is inconsistent (has a strange order).

In supplementary Fig. 10 caption there is a reference to "Fig. a" – I believe the number is missing.

p9 152 - How this ct/cc ratio value was obtained needs to be clarified. Was it from the O-CH₃ stretching of the DMC Raman spectra?

p32 578 - Energy cutoff for the AIMD - 500 Ry. Typo? (50 Ry?)

p7 101 - remove "a" and "conformer"

p8 139 - increasing,

p9 151 - cis-trans

p11 Fig 2 - switch 0.59 with 0.35 pie charts for clarity or typo?

p16 280 - NVT?

p5 75 - Minor - Fig c is referred to before b

p19 320 Fig S14 - Cannot distinguish the curves. A legend is needed for the colors used for M LiPF₆ EC/DEC electrolytes and ct/cc 0.82 electrolytes.

Reviewer #5

(Remarks to the Author)

Manuscript: NCOMMS-24-07587-A

Title: “Conformational isomerism breaks the electrolyte solubility limit and stabilizes 4.9 V Ni-rich layered cathodes”

We thank the editor and reviewers for carefully assessing our manuscript (NCOMMS-24-07587-A) and providing us the opportunity to further revise the manuscript in Nature Communications. We especially appreciate the insightful questions raised by the reviewers and welcome the opportunity to address these questions. The responses are listed point-by-point in the following contents, and revisions have been highlighted by **yellow color** in the revised manuscript. Following are our responses and detailed explanation towards these comments from the reviewers.

Responses to Reviewers:

To Reviewer #1:2-20

Response to Reviewers' comments

Reviewer #1:

General comments: The authors have addressed our concern (Nature Chemistry submission) by quantitatively measuring the presence of different conformational isomers of DMC. However, they have not confirmed the thesis for which they irreversibly change the isomerism from cis-cis to cis-trans at room temperature in presence of lithium ions. The DFT calculations of Figure 2a suggest that, in presence of lithium ions Li^+ , the cis-trans isomer is a lower minimum energy configuration than cis-cis, and therefore, being the energy barrier relatively low (4 kcal/mol \sim 16 kJ/mol) it can be achieved by thermal triggering. The NOESY NMR is able to confirm this. The data presented show that DMC, upon thermal triggering, can swap between cis-cis and cis-trans conformations, however they have not shown that the 17M solution in DMC presents univocally cis-trans isomers. We understand that the salt is not soluble at 17M in DMC at room temperature, but the authors claim that it is at higher temperature, and that the solution stays stable at room temperature because of the transition between cis-cis and cis-trans. We therefore suggest to perform the NOESY NMR of the 17M solution at high temperature, and then do the measurement, rigorously on the same sample, at RT again to show that the cis-trans is still the prevalent isomer.

The characterization reported here is still insufficient to prove their claims.

Response: Thank you very much for your comments and suggestions for further improving our manuscript. During the revision, a series of experiments and discussions have been added to provide more evidence to support our results and improve the quality of this manuscript. The NMR of pure DMC and the 17 M LiFSI-DMC electrolytes after returning to room temperature was performed to verify the role of Li^+ in configurational isomerization transitions. As suggested, the NOESY NMR of the 17 M electrolytes was conducted at high temperature and room temperature on the same sample. To exclude the presence of micro clustering and cation-anion recombination, the dynamic light scattering (DLS) test was supplemented. Furthermore, the ab initio molecular dynamics (AIMD) simulations were performed to analyze the effect of configurational isomerism on the solubility and solvation structures. Please also see the detailed responses to the following comments.

Comments 1: Response to the Referees 1 – “NOESY-NMR clearly identifies two isomers and temperature-driven interconversion.”

The two isomers are interchangeable with the increase of the temperature, and this was expected. The data provided do not show that in presence of lithium, one isomer prevails over the other, nor that is the reason for the increased solubility.

Response: Thanks so much for your valuable comments. In order to verify the effect of Li^+ on the solubility of different configurational isomers and the formed solvation structures, we conducted NMR tests on pure DMC solvent and 17 M LiFSI-DMC electrolytes after heating for 24 h at 90 °C. As shown in Supplementary Fig. 7a, only one single peak can be identified for the pure DMC solvent (without LiFSI salt) after heating for 24 h because H in cc-DMC has exactly the same chemical environment. But for the 17 M LiFSI-DMC electrolyte, two peaks appear since the ct-DMC is asymmetric with H in two different chemical environments. Note that, although the

DMC solvent and 17 M LiFSI-DMC electrolytes were heated at 90 °C for 24 h, the NMR test was performed after cooling to room temperature (298 K). Therefore, we can speculate that an important role of Li⁺ is to maintain the stability of the ct-DMC configuration isomer. When sufficient energy is input into the system, cc-DMC may transform into ct-DMC, whether it is pure DMC solvent or 17 M electrolyte. However, the Li⁺-ct-DMC is more stable than Li⁺-cc-DMC and free cc-DMC after introducing Li⁺, and the ct-DMC can still be preserved after cooling to room temperature. This is consistent with our previous DFT calculations (Fig. 2a and 2b). For pure DMC solvents, cc-DMC has lower energy than ct-DMC and is more stable. Therefore, ct-DMC would return to cc-DMC when cooling to room temperature, even if cc-DMC is converted to ct-DMC after heating. But when Li⁺ is introduced, Li⁺ coordinates with DMC, and Li⁺-ct-DMC has lower energy than Li⁺-cc-DMC. Therefore, the ct-DMC configuration generated by thermal triggering can still be maintained after the external excitation disappears.

To further study the impact of configurational isomerism on solubility, we performed ab initio molecular dynamics (AIMD) simulations for the 17 M LiFSI-cc-DMC and 17 M LiFSI-ct-DMC electrolytes. When the system reaches stability, the solvent and solute are not completely homogeneously mixed and aggregation of the solvation structure can be observed for the 17 M LiFSI-cc-DMC electrolytes, indicating that LiFSI cannot be completely dissolved in solvent that is entirely cc-DMC. But for the 17 M electrolytes with ct-DMC, the solvation structure is dispersed very evenly without obvious aggregation. More favorable evidence reflecting the effect of configurational isomerism on solubility is provided by radial distribution functions (RDFs) and coordination numbers. As shown in Fig. 4g, the sharp peak (1.85 Å) of Li⁺-O in DMC corresponding to the primary solvating shell, and the Li⁺-O in ct-DMC has higher intensity than the cc-DMC, demonstrating that the Li⁺ is more easily coordinated with ct-DMC than with cc-DMC. Correspondingly, the Li⁺-O in ct-DMC shows much larger coordination number than the Li⁺-O in cc-DMC (Fig. 4f). For the FSI⁻ anions, ct-DMC is also more easily coordinated with FSI⁻ than cc-DMC. The FSI⁻-O in ct-DMC shows much higher intensity than FSI⁻-O in cc-DMC for all observed peaks (Fig. 4h), and the FSI⁻-O in ct-DMC also exhibits much larger coordination number than the FSI⁻-O in cc-DMC (Fig. 4k). Therefore, both Li⁺ and FSI⁻ are more likely to coordinate with ct-DMC and enter its solvation shell than cc-DMC. The influence of solvent configuration isomerism on solubility and solvation structure can be attributed to two aspects. First, ct-DMC has smaller steric hindrance effect than cc-DMC when coordinated with Li⁺. Due to steric hindrance effect, only one O (C=O) in cc-DMC can directly coordinate with Li⁺ (Fig. 4b). But for the ct-DMC, both O (C=O) in ct-DMC can directly coordinate with Li⁺ due to the transformation to cis-trans isomerism, which weakens the steric hindrance effect. In addition, the transformation of cc-DMC to ct-DMC breaks the original mirror symmetry, causing ct-DMC to show greater polarity. Therefore, the ct-DMC has greater affinity for solutes and also exhibits higher solubility. We have supplemented the data and related discussions in the revised manuscript as follows:

Line 7-14, Page 12 “Then the ¹H NMR spectra of pure solvent and 17 M electrolytes were collected after cooling to room temperature (Supplementary Fig. 7a). Only one

single peak can be identified for the pure DMC solvent (without LiFSI salt) since H in cc-DMC has exactly the same chemical environment. However, two peaks can still be observed for the 17 M electrolytes because the ct-DMC is asymmetric with H in two different chemical environments, indicating that cc-DMC and ct-DMC coexist in the electrolytes. Therefore, we conclude that an important role of Li⁺ is to maintain the stability of the ct-DMC configuration isomer.”

Line 7-23, Page 15; Line 1-19, Page 16, “To better reveal the impact of configurational isomerism on the solubility and solvent structure, we conducted ab initio molecular dynamics (AIMD) simulations. For the 17 M electrolyte with cc-DMC, the solvent and solute are not completely homogeneously mixed and the aggregation of solvation structures can be observed (Fig. 4a), indicating that LiFSI cannot be completely dissolved in solvent that is entirely cc-DMC. In contrast, the 17 M electrolyte with ct-DMC, the solvation structure is dispersed very evenly without obvious aggregation (Fig. 4d). Apparently, LiFSI is more soluble in ct-DMC than in cc-DMC, which can be verified by radiation distribution functions (RDFs) and coordination numbers of the two electrolytes. As shown in Fig. 4g, the sharp peak (1.85 Å) of Li⁺-O in DMC corresponding to the primary solvating shell, and the Li⁺-O in ct-DMC has higher intensity than the cc-DMC, demonstrating that the Li⁺ is more easily coordinated with ct-DMC than cc-DMC. In addition, peaks corresponding to the second (3.00 Å) and third shells (3.91 Å) also have higher intensities for the ct-DMC. Correspondingly, the Li⁺-O in ct-DMC shows much larger coordination number than the Li⁺-O in cc-DMC (Fig. 4j). For the FSI⁻ anions, ct-DMC is also more easily coordinated with FSI⁻ than cc-DMC. The FSI⁻-O in ct-DMC shows much higher intensity than FSI⁻-O in cc-DMC for all observed peaks (Fig. 4h), and the FSI⁻-O in ct-DMC exhibits also much larger coordination number than the FSI⁻-O in cc-DMC (Fig. 4k). Therefore, both Li⁺ and FSI⁻ are more likely to coordinate with ct-DMC and enter its solvation shell compared with cc-DMC. The configuration of the solvent basically does not affect the association and dissociation between cations and anions (Fig. 4i and 4l). The difference in solvation structure of the electrolyte formed by cc-DMC and ct-DMC can be reflected by representative solvation structures. Due to steric hindrance effect, only one cc-DMC (the O in C=O) can directly coordinate with Li⁺ (Fig. 4b). But for the ct-DMC, two ct-DMC (the O in C=O) can directly coordinate with Li⁺ due to the formation of cis-trans isomerism, which weakens the steric hindrance effect. In addition, the transformation of cc-DMC to ct-DMC breaks the original mirror symmetry, causing ct-DMC to show greater polarity. Research conducted by Hyejin Lee et al. also indicates that the difference in polarity between configurational isomers is an important factor affecting salt dissociation⁴¹. Therefore, the ct-DMC has greater affinity for solutes and also exhibits higher solubility. Furthermore, interconversion between cc-DMC and ct-DMC was also observed in AIMD. At the beginning of the simulation, only cc-DMC (28) was given. However, there is a ct-DMC that appears after reaching stability (Fig. 4c). Similarly, one cc-DMC appears after reaching stability for the simulation with 28 ct-DMC molecules as the initial simulation conditions (Fig. 4f). It is consistent with the ¹H-¹H NOESY NMR spectrum (Fig. 3c and 3d). However, the conversion frequency of cc-DMC to ct-DMC is much higher than the conversion of ct-DMC to cc-DMC. The mutual

transformation between configurational isomers allows us to change the relative proportions of the two by manipulating conditions, thereby changing the property of the electrolyte.”

We sincerely hope our explanation can satisfy the reviewer on this question and are also sincerely looking forward to getting support from the reviewer.

Supplementary Fig. 7. a, ^1H spectra of 17 M electrolytes and DMC solvent at temperature after returning to 298 K.

Fig. 2 a, Transition state energies of different conformers of DMC solvent in the presence and absence of Li^+ . **b,** Schematic diagram of the interconversion of different conformers.

Fig.4 | AIMD simulation results. a, A snapshot of the simulation of 17 M LiFSI cc-DMC electrolytes. b, The representative solvation structure obtained from AIMD simulation. c, The solvation structures cycled in Fig. a and solvent configurations. d, A snapshot of the simulation of 17 M LiFSI ct-DMC electrolytes. e, The representative solvation structure obtained from AIMD simulation. f, The solvation structures cycled in Fig. d and solvent configurations. g-i, Radial distribution functions (RDFs) of Li-O in cc-DMC and ct-DMC (g), FSI-O in cc-DMC and ct-DMC (h), and Li-O in cc-DMC and ct-DMC (i). j-l, The coordination numbers of Li-O in cc-DMC and ct-DMC (j), FSI-O in cc-DMC and ct-DMC (k), and Li-O in cc-DMC and ct-DMC (l).

Comments 2: Page 7 line 13 – “The 1H NOESY spectra were collected at 333K and 363K.”

In order to prove irreversible isomer transformation, these measurements should include the RT after the thermal triggering.

Response: Many thanks for your helpful suggestions. We have conducted 1D ^1H and ^1H - ^1H NOESY spectra at room temperature after the thermal triggering as suggested. As shown in the Supplementary Fig. 7a, two peaks with different intensity can be observed after thermal triggering for the 1D spectra, indicating that cc-DMC and ct-DMC coexist, and this may be a system with exchanging. Then, ^1H - ^1H NOESY spectra

were collected at room temperature. At room temperature (298 K), four peaks can be observed, and peaks corresponding to the exchange of cis-cis (3.850 ppm) and cis-trans (3.862 ppm) were much weaker compared with the case tested at 333 K (Fig. 3d), indicating that the exchanging between cis-cis and cis-trans was reduced with the decrease of temperature. Therefore, conversions between configurational isomers can be confirmed. Related discussion has been supplemented in the revised manuscript as follows:

Line 7-19, Page 12 “Then the ^1H NMR spectra of pure solvent and 17 M electrolytes were collected after cooling to room temperature (Supplementary Fig. 7a). Only one single peak can be identified for the pure DMC solvent (without LiFSI salt) since H in cc-DMC has exactly the same chemical environment. However, two peaks can still be observed for the 17 M electrolytes because the ct-DMC is asymmetric with H in two different chemical environments, indicating that cc-DMC and ct-DMC coexist in the electrolytes. Therefore, we conclude that an important role of Li^+ is to maintain the stability of the ct-DMC configuration isomer. And then, ^1H - ^1H NOESY spectra were collected at room temperature (298 K). As shown in Supplementary Fig. 7b, four peaks can be observed, and peaks corresponding to the exchange of cis-cis (3.850 ppm) and cis-trans (3.862 ppm) were much weaker compared with the case tested at 333 K (Fig. 3d), indicating that cis-trans configuration is preserved in the presence of Li^+ after cooling, and exchanging between cis-cis and cis-trans was reduced with the decrease of temperature.”

Supplementary Fig. 7. a, ^1H spectra of 17 M electrolytes and DMC solvent at temperature after returning to 298 K. b, The fragment of ^1H - ^1H NOESY NMR spectrum of 17 M electrolytes after returning to 298 K.

Comment 3: Page 7 line 7 – “Interestingly, even after standing at room temperature for a week, the electrolyte remains clear and transparent without salt precipitation (Fig. 1c).”

Note there are two Fig. 1c.

By comparison with the 1M – 13 M solutions, the 17 M looks opaquer. Due to the high concentration of salt, this is to be expected, however we advise DLS or SAXS/WAXS measurements to exclude the presence of micro clustering and cation-anion recombination, that the authors confirm will occur as stated by the response to comment 4.

Response: Thanks so much for the helpful comments. Compared with 1 M – 13 M solutions, the 17 M electrolyte looks opaquer, which may be caused by the influence of the shooting angle or lighting. We photographed the 17 M electrolyte alone to show the visual effect. The presence of micro clustering and cation-anion recombination may be visually imperceptible. Therefore, we further conducted DLS testing (Supplementary Fig. 1) according to the reviewer's suggestions. For the 17 M electrolytes after heating for 24 h, no cluster was observed over the entire test range (0.2-5000 nm), indicating that the 17 M electrolytes are homogeneous solutions rather than colloids or suspensions after adequate heating. As a comparison, we also conducted DLS on the 17 M electrolytes after heating for 1 h. As shown in Supplementary Fig. 1, a peak at about 260 nm can be identified, suggesting that the LiFSI salt precipitates during the cooling and aging processes. Therefore, heating time is an important factor to promote the conformational transition and ensure that it forms a stable solution. We have supplemented related data and discussions in the revised manuscript as follows:

Line 10-13, Page 7 “No nanoclusters were detected by dynamic light scattering (DLS), indicating that the 17 M electrolytes are homogeneous solutions rather than colloids or suspensions after adequate heating (Supplementary Fig. 1). But for the 17 M after heating for only 1 h, a peak at about 260 nm can be identified, suggesting that the LiFSI salt precipitates during the cooling and aging processes.”

Line 9-11, Page 29 “The DLS was conducted on an OMEC NS-90Z Nanoparticle Size and Zeta Potential Analyzer using malvern cuvettes (DTS0012)”

Supplementary Fig. 1. Dynamic light scattering (DLS) analysis for the 17 M LiFSI-DMC electrolytes after heating 24 h and 1 h.

Comment 4: Page 8 line 8 – “With the increase of concentration, the intensity of free

cis-cis DMC located at 918 cm⁻¹ gradually decreases (Fig. 2d) after exceeding 1 M, while the bonded cis-cis DMC, cis-trans DMC configuration gradually increases.” The Raman data and the authors confirm that there is a significant portion of cc-DMC that is interacting with the salt namely ‘bound cc-DMC’. This excludes than the isomeric transition to be the only reason for the increased solubility.

Response: Thanks so much for the constructive comments. For both aqueous and non-aqueous electrolytes, the free solvent gradually decreases and the bonded-solvent gradually increases with increasing salt concentration (*Nat Energy* **2019**, *4*, 269-280). For the LiFSI-DMC electrolyte system, there are two solvent configurations at the same time and thus there are two types of bonded-solvent forms. According to our previous DFT calculations, the Gibbs free energy of the transformation of cc-DMC to Li⁺-ct-DMC is negative, so this process is thermodynamically favorable, which is consistent with previous reports (*J. Phys. Chem. Lett.* **2020**, *11*, 10382-10387). Therefore, the coordinated solvent configuration in any concentration of electrolytes should be bonded ct-DMC. However, the experimental results are indeed that both cc-DMC and ct-DMC exist in the electrolyte at any concentration. We speculate that this may be due to the presence of high-energy intermediate species. As shown in Supplementary Fig. 2, the Gibbs free energy of the transition from cc-DMC to Li⁺-ct-DMC is negative, but it needs to undergo the transition state of high-energy ct-DMC. Therefore, the electrolyte always contains both cc-DMC and ct-DMC. After heating, the energy of cc-DMC increases, allowing it to cross the transition barrier to ct-DMC and then transform into Li⁺-ct-DMC. This speculation can also be verified from AIMD simulations. The interconversion between cc-DMC and ct-DMC was observed in AIMD. At the beginning of the simulation, only cc-DMC (28) was given. However, there is a ct-DMC that appears after reaching stability (Fig. 4c). Similarly, one cc-DMC appears after reaching stability for the simulation with 28 ct-DMC molecules as the initial simulation condition (Fig. 4f). However, the conversion frequency of cc-DMC to ct-DMC is much higher than the conversion of ct-DMC to cc-DMC. This is consistent with our previous explanation, that is, the barrier for the transition from Li⁺-ct-DMC to ct-DMC is much larger than the transition from cc-DMC to ct-DMC. Therefore, there is a certain proportion of bonded cc-DMC in the highly concentrated electrolyte. Moreover, increasing the temperature can transform bonded cc-DMC into bonded ct-DMC. We sincerely thank the reviewers for their valuable comments that enabled us to have a more in-depth analysis and discussion of the property of the electrolyte. We have added related discussion in the revised manuscript as follows:

Line 16-23, Page 8 “However, according to DFT calculations, the Gibbs free energy of the transformation of cc-DMC to Li⁺-ct-DMC is negative, and this process is thermodynamically favorable, which is consistent with previous reports⁴¹. Therefore, the coordinated solvation configuration in any concentration of electrolytes should be bonded ct-DMC. It is contrary to the Raman results. We speculate that this may be due to the presence of high-energy intermediate species. As shown in Supplementary Fig. 2, the free energy of the transition from cc-DMC to Li⁺-ct-DMC is negative, but it needs to undergo the transition state of high-energy ct-DMC. Therefore, the electrolyte always contains both cc-DMC and ct-DMC.”

Line 11-19, Page 16 “Furthermore, interconversion between cc-DMC and ct-DMC was also observed in AIMD. At the beginning of the simulation, only cc-DMC (28) was given. However, there is a ct-DMC that appears after reaching stability (Fig. 4c). Similarly, one cc-DMC appears after reaching stability for the simulation with 28 ct-DMC molecules as the initial simulation conditions (Fig. 4f). It is consistent with the ¹H-¹H NOESY NMR spectrum (Fig. 3c and 3d). However, the conversion frequency of cc-DMC to ct-DMC is much higher than the conversion of ct-DMC to cc-DMC. The mutual transformation between configurational isomers allows us to change the relative proportions of the two by manipulating conditions, thereby changing the property of the electrolyte.”

Supplementary Fig. 2. Reaction coordinate for the transformation of cc-DMC to Li⁺-ct-DMC.

Fig.4 a, A snapshot of the simulation of 17 M LiFSI cc-DMC electrolytes. c. The solvation structures cycled in Fig. a and solvent configurations. d, A snapshot of the simulation of 17 M LiFSI ct-DMC electrolytes. f. The solvation structures cycled in Fig. d and solvent configurations.

Comment 5: Page 9 line 6 – “In this case, all solvation structures are anion-mediated, and the solvation structures are cross-linked to each other via Li⁺ or FSI⁻ to form a polymer-like three-dimensional structure.”

Response to Referee 9 – The authors in the response to the referee stated: “The polymer-like three-dimensional structure we describe here is not a real polymer structure.”

We agree with the authors and we believe the text needs to be rephrased because the characterisation provided is insufficient to confirm or exclude the solvation structure proposed.

Response to Referee 9 – The structure we describe here is a three-dimensional interconnected structure formed by the tight combination of FSI⁻ and Li⁺. Without further analysis the proposed structure cannot be confirmed as, at such concentrations, the solution can easily be microscopically inhomogeneous. We believe that the analysis does not support the result and that there is no evidence of the proposed solvation structure.

Response: Thanks for your comments and suggestions. We agree with the reviewer that using a three-dimensional interconnected structure to describe the electrolyte is not accurate enough. In order to describe this solvent structure more accurately, we performed AIMD simulations. For the 17 M LiFSI-cc-DMC electrolyte, only one cc-DMC molecule and two FSI⁻ anions can enter the first solvation shell of Li⁺. For the 17 M LiFSI-ct-DMC electrolyte, two ct-DMC molecules and two FSI⁻ anions can enter the first solvation shell of Li⁺. For the actual 17 M electrolyte containing both cc-DMC and ct-DMC, its solvent configuration is closer to that shown in Fig. 4c and 4f. One cc-DMC and one ct-DMC, two FSI⁻ anions surround the Li⁺ to form the first solvation shell. The solvents and anions in the first shell are often unique to the central Li⁺, but the second or even third outer shells are often shared with other Li⁺ as shown in Fig. 4a and 4d. To make it clearer, we have rephrased the sentence and supplemented related discussion in the revised manuscript as follows:

Line 19-23, Page 16; Line 1, Page 17 “For the actual 17 M electrolyte containing both cc-DMC and ct-DMC, its solvation configuration is between 17 M LiFSI-cc-DMC and 17 M LiFSI-ct-DMC, which is similar to the structure shown in Fig. 4c and 4f. One cc-DMC and one ct-DMC and two FSI⁻ anions surround the Li⁺ to form the first solvation shell. The solvents and anions in the first shell are often unique to the central Li⁺, but the solvents or anions in the second or even third outer shells are often shared with other Li⁺ as shown in Fig. 4a and 4d.”

Fig.4 a, A snapshot of the simulation of 17 M LiFSI cc-DMC electrolytes. b, The representative solvation structure obtained from AIMD simulation. c, The solvation structures cycled in Fig. a. d, A snapshot of the simulation of 17 M LiFSI ct-DMC electrolytes. e, The representative solvation structure obtained from AIMD simulation. f, The solvation structures cycled in Fig. d.

Comment 6: Page 12 line 3 – “Combining spectroscopic analysis and theoretical calculations, the formation of highly concentrated electrolytes derived from thermally induced conformers transitions can be clearly interpreted. The increase in temperature can promote the transformation of cis-cis to cis-trans, and the highly polar cis-trans conformer has higher solubility compared to the cis-cis configuration. “ The statement that the cis-trans has higher solubility than cis-cis is still unsupported. This needs a reference or solubility measurements. The change in isomorphism could be one, but not the only, reason of the increased solubility.

Response: Thanks so much for your valuable comments. Experimentally, we cannot prepare electrolytes containing only cc-DMC or ct-DMC since cc-DMC and ct-DMC always coexist in the electrolyte. However, for theoretical simulations, we can study the impact of different configurational isomers on solubility by artificially setting relevant parameters. For the initial system, all solvent molecules were completely configured as cc-DMC or ct-DMC. By comparing the radial distribution functions (RDFs) and the coordination numbers of Li^+ after the system reaches stability, we can quantitatively compare the difference in solubility of LiFSI in cc-DMC and ct-DMC. For the 17 M electrolyte with cc-DMC, the solvent and solute are not completely homogeneously mixed and the aggregation of the solvation structure can be observed, indicating that LiFSI cannot be completely dissolved in cc-DMC solvent. In contrast, the 17 M electrolyte with ct-DMC, the solvation structure is dispersed very evenly without obvious aggregation. Therefore, LiFSI is more soluble in ct-DMC than in cc-DMC, which can be verified by radiation distribution function and coordination number of the two electrolytes. As shown in Fig. 4g, the sharp peak (1.85 Å) of Li^+ -O in DMC corresponding to the primary solvating shell, and the Li^+ -O in ct-DMC has higher

intensity than the cc-DMC, demonstrating that the Li^+ is more easily coordinated with ct-DMC than with cc-DMC. In addition, the peaks corresponding to the second (3.00 Å) and third shells (3.91 Å) also have higher intensities for the ct-DMC. Correspondingly, the $\text{Li}^+\text{-O}$ in ct-DMC shows much larger coordination number than the $\text{Li}^+\text{-O}$ in cc-DMC (Fig. 4f). For the FSI^- anions, c-t DMC is also more easily coordinated with FSI^- than cc-DMC. The $\text{FSI}^-\text{-O}$ in c-t DMC shows much higher intensity than $\text{FSI}^-\text{-O}$ in cc-DMC for all observed peaks (Fig. 4h), and the $\text{FSI}^-\text{-O}$ in ct-DMC exhibits also much larger coordination number than the $\text{FSI}^-\text{-O}$ in cc-DMC (Fig. 4k). Therefore, both Li^+ and FSI^- are more likely to coordinate with ct-DMC and enter its coordination shell than cc-DMC. The configuration of the solvent basically does not affect the association and dissociation between cations and anions (Fig. 4i and 4l). The difference in solvation structures of the electrolyte formed by cc-DMC and ct-DMC can be reflected by representative solvation structures. Due to steric hindrance effect, only one O (C=O) in cc-DMC can directly coordinate with Li^+ (Fig. 4b). But for the ct-DMC, both O (C=O) in ct-DMC can directly coordinate with Li^+ due to the transformation to cis-trans isomerism, which weakens the steric hindrance effect. In addition, the transformation of cc-DMC to ct-DMC breaks the original mirror symmetry, causing c-t DMC to show greater polarity. Therefore, the ct-DMC has greater affinity for solutes and also exhibits greater solubility. The research conducted by Hyejin Lee et al. also shows that ct-DMC has greater polarity than cc-DMC, which is more conducive to dissociating Li salts and thus has higher solubility (*J. Phys. Chem. Lett.* **2020**, *11*, 10382-10387). We have added related discussions and simulation details in the revised manuscript as follows:

Line 7-23, Page 7; Line 1-11, Page 8 “To better reveal the impact of configurational isomerism on the solubility and solvent structure, we conducted ab initio molecular dynamics (AIMD) simulations. For the 17 M electrolyte with cc-DMC, the solvent and solute are not completely homogeneously mixed and the aggregation of solvation structures can be observed (Fig. 4a), indicating that LiFSI cannot be completely dissolved in solvent that is entirely cc-DMC. In contrast, the 17 M electrolyte with ct-DMC, the solvation structure is dispersed very evenly without obvious aggregation (Fig. 4d). Apparently, LiFSI is more soluble in ct-DMC than in cc-DMC, which can be verified by radiation distribution functions (RDFs) and coordination numbers of the two electrolytes. As shown in Fig. 4g, the sharp peak (1.85 Å) of $\text{Li}^+\text{-O}$ in DMC corresponding to the primary solvating shell, and the $\text{Li}^+\text{-O}$ in ct-DMC has higher intensity than the cc-DMC, demonstrating that the Li^+ is more easily coordinated with ct-DMC than cc-DMC. In addition, peaks corresponding to the second (3.00 Å) and third shells (3.91 Å) also have higher intensities for the ct-DMC. Correspondingly, the $\text{Li}^+\text{-O}$ in ct-DMC shows much larger coordination number than the $\text{Li}^+\text{-O}$ in cc-DMC (Fig. 4j). For the FSI^- anions, ct-DMC is also more easily coordinated with FSI^- than cc-DMC. The $\text{FSI}^-\text{-O}$ in ct-DMC shows much higher intensity than $\text{FSI}^-\text{-O}$ in cc-DMC for all observed peaks (Fig. 4h), and the $\text{FSI}^-\text{-O}$ in ct-DMC exhibits also much larger coordination number than the $\text{FSI}^-\text{-O}$ in cc-DMC (Fig. 4k). Therefore, both Li^+ and FSI^- are more likely to coordinate with ct-DMC and enter its solvation shell compared with cc-DMC. The configuration of the solvent basically does not affect the association and

dissociation between cations and anions (Fig. 4i and 4l). The difference in solvation structure of the electrolyte formed by cc-DMC and ct-DMC can be reflected by representative solvation structures. Due to steric hindrance effect, only one cc-DMC (the O in C=O) can directly coordinate with Li⁺ (Fig. 4b). But for the ct-DMC, two ct-DMC (the O in C=O) can directly coordinate with Li⁺ due to the formation of cis-trans isomerism, which weakens the steric hindrance effect. In addition, the transformation of cc-DMC to ct-DMC breaks the original mirror symmetry, causing ct-DMC to show greater polarity. Research conducted by Hyejin Lee et al. also indicates that the difference in polarity between configurational isomers is an important factor affecting salt dissociation⁴¹. Therefore, the ct-DMC has greater affinity for solutes and also exhibits higher solubility.”

Line 19-23, Page 29; Line 1-6, Page 30 “To reveal the impact of configurational isomerism on the solubility and solvation structures, the CP2K/QUICKSTEP package was used for NPT simulations at T=300K^{61,62}. Density functional theory based on hybrid Gaussian plane wave (GPW) approach and periodic boundary conditions (PBC) was implemented. The Perdew-Burke-Ernzerhof (PBE) exchange and correlation functional⁶³ as well as the dispersion correction with the Grimme D3 method⁶⁴ was employed. The Goedecker-Teter-Hutter pseudopotentials⁶⁵ and the optimized short range double zeta basis set with a polarization function (DZVP-MOLOPT-SR)⁶⁶ were performed for the calculations. The energy for the cutoff was set to 500 Ry. The initial two models of a cubic box of 35 Å were built with packmol package^{67,68}, where contained 28 ct-DMC/cc-DMC and 40 LiFSI. The pressure was maintained under 1 atm during the simulation. The dynamical simulation was run for 50000 steps in the NPT ensemble with a timestep of 1 fs.”

Reference

61. VandeVondele, J. et al. Quickstep: Fast and accurate density functional calculations using a mixed Gaussian and plane waves approach. *Comput. Phys. Commun.* **167**, 103-128 (2005).
62. Kühne, T. D. et al. CP2K: An electronic structure and molecular dynamics software package-Quickstep: Efficient and accurate electronic structure calculations. *J. Chem. Phys.* **152** (2020).
63. Perdew, J. P., Burke, K. & Ernzerhof, M. Generalized gradient approximation made simple. *Phys. Rev. Lett.* **77**, 3865 (1996).
64. Grimme, S., Antony, J., Ehrlich, S. & Krieg, H. A consistent and accurate ab initio parametrization of density functional dispersion correction (DFT-D) for the 94 elements H-Pu. *J. Chem. Phys.* **132** (2010).
65. Krack, M. Pseudopotentials for H to Kr optimized for gradient-corrected exchange-correlation functionals. *Theor Chem Acc.* **114**, 145-152 (2005).
66. VandeVondele, J. & Hutter, J. Gaussian basis sets for accurate calculations on molecular systems in gas and condensed phases. *J. Chem. Phys.* **127** (2007).
67. Martínez, Leandro, et al. PACKMOL: A package for building initial configurations for molecular dynamics simulations. *J. Comput. Chem.* **30**, 2157-2164 (2009).
68. Martínez, José Mario, and Leandro Martínez. Packing optimization for automated generation of complex system's initial configurations for molecular dynamics and

Fig.4 | AIMD simulation results. a, A snapshot of the simulation of 17 M LiFSI cc-DMC electrolytes. b, Representative solvation structures from AIMD simulation. c, The solvation structures cycled in Fig. a and solvent configurations. d, A snapshot of the simulation of 17 M LiFSI ct-DMC electrolytes. e, Representative solvation structures from AIMD simulation. f, The solvation structures cycled in Fig. d and solvent configurations. g-i, Radial distribution functions (RDFs) of Li-O in cc-DMC and ct-DMC (g), FSI-O in cc-DMC and ct-DMC (h), and Li-O in cc-DMC and ct-DMC (i). j-l, The coordination numbers of Li-O in cc-DMC and ct-DMC (j), FSI-O in cc-DMC and ct-DMC (k), and Li-O in cc-DMC and ct-DMC (l).

Comment 7: Page 12 line 7 – “In addition, the cis-trans conformer has lower energy than the equivalent cis-cis conformer after coordinating with Li⁺. Therefore, the unique electrolyte configuration of the electrolyte obtained at high temperatures can be preserved even after cooling, which enables its application in Li-ion batteries at room temperature.” The spectroscopic measurements do not support this point as they do not show that in the presence of Li⁺ the cis-trans configuration is preserved after cooling. The NOESY shows an interchange between cis-trans as a function of the temperature, as expected.

Response: Thanks so much for your valuable comments. We have collected the 1D

^1H and ^1H - ^1H NOESY NMR spectrum after cooling. As shown in the Supplementary Fig. 7a, only one peak can be observed for the pure DMC solvent (without LiFSI salt) after heating for 24 h because H in cc-DMC has the same chemical environment. However, two peaks can be identified after cooling, indicating that ct-DMC appears, and cc-DMC and ct-DMC coexist in the electrolytes. Obviously, in the presence of Li^+ , the cis-trans configuration is preserved after cooling. And then, ^1H NOESY spectra were collected at room temperature (298 K). As shown in Supplementary Fig. 7b, four peaks can be observed, and peaks corresponding to the exchange of cis-cis (3.850 ppm) and cis-trans (3.862 ppm) were much weaker compared with the case tested at 333 K (Fig. 3d), indicating that cis-trans configuration is preserved in the presence of Li^+ after cooling, and exchanging between cis-cis and cis-trans was reduced with the decrease of temperature. In the presence of Li^+ , ct-DMC can not only be preserved, but there is also an exchange between cc-DMC and ct-DMC. We have added related discussion in the revised manuscript as follows:

Line 7-19, Page 12 “Then the ^1H NMR spectra of pure solvent and 17 M electrolytes were collected after cooling to room temperature (Supplementary Fig. 7a). Only one single peak can be identified for the pure DMC solvent (without LiFSI salt) since H in cc-DMC has exactly the same chemical environment. However, two peaks can still be observed for the 17 M electrolytes because the ct-DMC is asymmetric with H in two different chemical environments, indicating that cc-DMC and ct-DMC coexist in the electrolytes. Therefore, we conclude that an important role of Li^+ is to maintain the stability of the ct-DMC configuration isomer. And then, ^1H - ^1H NOESY spectra were collected at room temperature (298 K). As shown in Supplementary Fig. 7b, four peaks can be observed, and peaks corresponding to the exchange of cis-cis (3.850 ppm) and cis-trans (3.862 ppm) were much weaker compared with the case tested at 333 K (Fig. 3d), indicating that cis-trans configuration is preserved in the presence of Li^+ after cooling, and exchanging between cis-cis and cis-trans was reduced with the decrease of temperature.”

Supplementary Fig. 7. a, ^1H spectra of 17 M electrolytes and DMC solvent at temperature after returning to 298 K. b, The fragment of ^1H - ^1H NOESY NMR spectrum of 17 M electrolytes after returning to 298 K.

Comment 8: Page 10 Figure 2e – We believe that the fit to the Raman data should be presented here. It is not clear what the dotted lines are indicating (spectra regions?) as they do not correspond to any peak position. This point can be addressed by expanding the caption. The ratio between ct and cc needs to be quantitatively evaluated. In the highest regime ct/cc is still < 1 , which means that there is still a greater ratio of cc than ct.

Response: Thanks so much for your helpful comments and suggestions. The dotted lines in Fig. 2c and 2e represent the corresponding peak position to the solvation configuration/solvent form. For example, dotted lines at 720 cm^{-1} correspond to the peak position of the SSIP in the fitting. In order to display the Raman results more clearly, the fitting results of Raman data have been presented in Fig. 2c and 2e. In addition, we have also explained it in the Figure caption. The ratio between ct and cc can be quantitatively evaluated from the fitting results of O-CH₃ stretching of the DMC. As shown in Figure 2c, free cc-DMC, coordinated cc-DMC, free ct-DMC and coordinated ct-DMC were fitted respectively and the relative proportions of the corresponding components were labeled. As the reviewer commented, even in the

highest regime, ct/cc is still < 1 . There is still a greater ratio of cc than ct. We understand the reviewer's doubt as to why there is a large proportion of cc-DMC in the electrolyte. This is a very important and valuable question to understand the solvation configuration and solvent structure in this highly concentrated electrolyte, which was also proposed in comment 4. We would like to explain again and clear the reviewer's confusion. First, experiments (Raman and NMR) confirm that there are two configurations of DMC in the LiFSI-DMC electrolyte system, namely cc-DMC and ct-DMC. However, according to the DFT calculations, Li⁺-ct-DMC is thermodynamically more stable than cc-DMC. Therefore, the electrolyte should only contain the Li⁺-ct-DMC without the presence of cc-DMC. We speculate that this may be due to the presence of high-energy intermediate species. The Gibbs free energy of the transition from cc-DMC to Li⁺-ct-DMC is negative. However, it needs to undergo the transition state of high-energy ct-DMC. Therefore, the electrolyte always contains both cc-DMC and ct-DMC, which can also be verified from AIMD simulations. The interconversion between cc-DMC and ct-DMC was observed in AIMD. At the beginning of the simulation, only cc-DMC (28) was given. However, there is a ct-DMC that appears after reaching stability (Fig. 4c). Similarly, one cc-DMC appears for the simulation with 28 ct-DMC molecules as the initial simulation conditions after reaching stability (Fig. 4f). However, the conversion frequency of cc-DMC to ct-DMC is much higher than the conversion of ct-DMC to cc-DMC. This is consistent with our previous explanation, that is, the potential barrier for the transition from Li⁺-ct-DMC to ct-DMC is much larger than the transition from cc-DMC to ct-DMC. Therefore, there is a certain proportion of bound cc-DMC in the highly concentrated electrolyte. We have added related discussion in the revised manuscript as follows:

Line 16-23, Page 8 “However, according to DFT calculations, the Gibbs free energy of the transformation of cc-DMC to Li⁺-ct-DMC is negative, and this process is thermodynamically favorable, which is consistent with previous reports⁴¹. Therefore, the coordinated solvation configuration in any concentration of electrolytes should be bonded ct-DMC. It is contrary to the Raman results. We speculate that this may be due to the presence of high-energy intermediate species. As shown in Supplementary Fig. 2, the free energy of the transition from cc-DMC to Li⁺-ct-DMC is negative, but it needs to undergo the transition state of high-energy ct-DMC. Therefore, the electrolyte always contains both cc-DMC and ct-DMC.”

Line 11-19, Page 16 “Furthermore, interconversion between cc-DMC and ct-DMC was also observed in AIMD. At the beginning of the simulation, only cc-DMC (28) was given. However, there is a ct-DMC that appears after reaching stability (Fig. 4c). Similarly, one cc-DMC appears after reaching stability for the simulation with 28 ct-DMC molecules as the initial simulation conditions (Fig. 4f). It is consistent with the ¹H-¹H NOESY NMR spectrum (Fig. 3c and 3d). However, the conversion frequency of cc-DMC to ct-DMC is much higher than the conversion of ct-DMC to cc-DMC. The mutual transformation between configurational isomers allows us to change the relative proportions of the two by manipulating conditions, thereby changing the property of the electrolyte.”

We sincerely hope our explanation can satisfy the reviewer on this question and

are also sincerely looking forward to getting support from the reviewer.

Fig. 2 c, The fitting results of Raman spectra (O-CH₃ stretching of the DMC) of various electrolytes. **e**, The fitting results of Raman spectra (S-N stretching of the FSI) of various electrolytes. The positions of the dashed lines and arrows in Fig. c and e correspond to the peak positions of the relevant solvation structures.

Supplementary Fig. 2. Reaction coordinate for the transformation of cc-DMC to Li⁺-ct-DMC.

Fig.4 a, A snapshot of the simulation of 17 M LiFSI cc-DMC electrolytes. c. The solvation structures cycled in Fig. a and solvent configurations. d, A snapshot of the simulation of 17 M LiFSI ct-DMC electrolytes. f. The solvation structures cycled in Fig. d and solvent configurations.

Manuscript: NCOMMS-24-07587-A

Title: “Conformational isomerism breaks the electrolyte solubility limit and stabilizes 4.9 V Ni-rich layered cathodes”

We thank the editor and reviewers for carefully assessing our manuscript (NCOMMS-24-07587-A) and providing us the opportunity to further revise the manuscript in Nature Communications. We especially appreciate the insightful questions raised by the reviewers and welcome the opportunity to address these questions. The responses are listed point-by-point in the following contents, and revisions have been highlighted by **yellow color** in the revised manuscript. Following are our responses and detailed explanation towards these comments from the reviewers.

Responses to Reviewers:

To Reviewer #1:2-3

To Reviewer #3:4-12

Response to Reviewers' comments

Reviewer #1:

General comments: The authors have now addressed the comments satisfactorily. I recommend the article for publication. I have a few minor comments that could help further improve the manuscript:

Response: Thanks so much to the reviewers for supporting the publication of our article. We are also very grateful for the reviewers' comments and suggestions for further improving our manuscript. The manuscript was carefully revised based on the reviewers' comments. The detailed responses to the comments are shown below.

Comments 1: 'As shown in Fig. 4g, the sharp peak (1.85 Å) of Li⁺-O in DMC corresponding to the primary solvating shell, and the Li⁺-O in ct-DMC has higher intensity than the cc-DMC, demonstrating that the Li⁺ is more easily coordinated with ct-DMC than with cc-DMC.'

• We note that increased intensity does not always mean higher coordination as the coordination is the integral of the peaks. We would advice to say 'suggesting' rather than 'demonstrating'.

Response: Thanks so much for your valuable comments. We have to admit that the reviewer's comment is correct that increased intensity does not always mean higher coordination as the coordination is the integral of the peaks. The manuscript has been revised combined with the supplemented AIMD simulation results as follows:

Line 8-11, Page 15 "As shown in Fig. 4g, the sharp peak (1.85 Å) of Li⁺-O in DMC corresponding to the primary solvating shell, and the Li⁺-O in ct-DMC has higher intensity than the cc-DMC, suggesting that the Li⁺ is more easily coordinated with ct-DMC than cc-DMC."

Comments 2: Radial not radiation distribution function

Response: Many thanks for your reminding. We have revised manuscript as follows:

Line 7-8, Page 15 "Apparently, LiFSI is more soluble in ct-DMC than in cc-DMC, which can be verified by radial distribution functions (RDFs) and coordination numbers of the two electrolytes."

Comment 3: We believe that the response to Comment 4 could be rephrased more positively, and we do not think that the DFT calculations and the Raman data are contradicting each other. The transition from cc to ct is confirmed by the NMR and both isomers exist in solution. We believe that it is one reason, but not the only reason for increased solubility. Another reason as the authors mention is the reduction of steric hindrance. We therefore suggest not to mention intermediates that are not consistent with the data showed.

Response: Thanks so much for the helpful comments, which enables us to think more deeply about the effect of configurational isomerization on solubility. We agree with the reviewer that the DFT calculations and the Raman data are not contradicting each other. As the concentration increases, the proportion of solvation structures related to ct-DMC increases, which is consistent with the Raman results since the transformation of Li⁺-cc-DMC to Li⁺-ct-DMC is thermodynamically favorable. Regarding the response to Comment 4, we agree with the reviewer that the transition from cc-DMC to ct-DMC

is one reason for the increased solubility, which has been confirmed by the NMR. Another reason is the reduction of steric hindrance, which was discussed in previous revised manuscript. The existence of high-energy intermediate species is our speculation, and it is not appropriate to use this to explain the effect on solubility. We have deleted the relevant discussion in the revised manuscript.

We sincerely hope our explanation can satisfy the reviewer on these questions and are also sincerely looking forward to getting support from the reviewer.

Reviewer #3:

General comments: This work proposed a conformational isomerism mechanism to break the electrolyte solubility limit and stabilize 4.9 V Ni-rich layered cathodes. The solvation structures were comprehensively probed by both experimental and theoretical characterizations. Further, excellent electrochemical performances were obtained due to the super-high salt concentration. I go through both the revised manuscript and rebuttal materials and personally think this work can hardly meet the high standards of Nature Communications. Details reasons are provided as follows.

One critical point is that whether the cis-trans solvent structures exist at room temperature. A lot of experimental characterizations including NMR were conducted to probe this point. However, there are several major issues for the simulation results.

Response: Thank you very much for your comments and suggestions for further improving our manuscript. During the revision, a series of theoretical calculations and simulations have been added to provide more evidence to support our results and improve the quality of this manuscript. We carefully checked the parameters of theoretical calculations and simulations to ensure the reliability of the results. Please also see the detailed responses to the following comments.

Comments 1: Figure 2a is used to explain the energy difference of ct and cc structures. However, the influence of entropy and temperature is not included in the total energy. If ct structure is always more stable than cc structure when coordinating with a Li ion, the calculation results can not explain the different solvation structures induced by different heating times.

Response: Thanks so much for your valuable comments. The reviewer's comments are crucial to understanding the transformation of cc-DMC to ct-DMC and why prolonged heating time promotes this transformation. If ct-DMC is always more stable than cc-DMC when coordinating with a Li⁺, the solvent coordinated with Li⁺ should be dominated by ct-DMC and is not affected by heating time as commented by the reviewer. To answer this question, we calculated the Gibbs free energy difference of Li⁺-cc-DMC and Li⁺-ct-DMC structures and the Gibbs free energy of the transition state after taking into account the effects of entropy and temperature as shown in Fig. 2a. It is well known that an edge-CH₃ in cc-DMC can be transformed into ct-DMC by rotating 180° along the adjacent C-O bond. When -CH₃ rotates about 90° along the C-O bond, the energy reaches its maximum value, which is the transition state energy we are searching for. In this case, the kinetic barrier for the transformation of Li⁺-cc-DMC to Li⁺-ct-DMC is 8.077 kcal/mol, and that for the transformation of Li⁺-ct-DMC to Li⁺-cc-DMC is 12.174 kcal/mol (Fig. 2a). Actually, kinetic energy difference to cross the transition state leads to the difference in transformation kinetics between cc-DMC and ct-DMC and their coexistence at equilibrium. The thermodynamic barrier (4.096 kcal/mol) determines the ratio of Li⁺-cc-DMC to Li⁺-ct-DMC. The thermodynamic barrier increases from 4.096 kcal/mol to 4.194 kcal/mol as the temperature increases from 298 K to 363 K (Supplementary Fig. 3), which is favorable to increase the proportion of ct-DMC. Therefore, increasing the temperature or increasing the heating time will change the ratio of Li⁺-cc-DMC to Li⁺-ct-DMC. We sincerely thank the reviewer's suggestion which enables us to clarify this issue, and we have also added related

discussion in the revised manuscript as follows:

Line 20-23, Page 7; Line 1-12, Page 8 “However, the cis-trans conformer becomes the most stable structure when taking the effect of Li^+ into consideration for the real electrolytes (Fig. 2a). The Gibbs free energy of Li^+ -ct-DMC is 4.096 kcal/mol lower than that of Li^+ -cc-DMC. However, the transition from Li^+ -cc-DMC to Li^+ -ct-DMC requires crossing a high energy transition state. It is well known that an edge- CH_3 in cc-DMC can be transformed into ct-DMC by rotating 180° along the adjacent C-O bond. The energy reaches its maximum value as the $-\text{CH}_3$ rotates about 90° along the C-O bond, which is the transition state shown in Fig. 2a. In this case, the kinetic energy barrier for the transformation of Li^+ -cc-DMC to Li^+ -ct-DMC is 8.077 kcal/mol, and the kinetic energy barrier for the transformation of Li^+ -ct-DMC to Li^+ -cc-DMC is 12.174 kcal/mol. Actually, kinetic energy difference to cross the transition state leads to the difference in transformation kinetics between cc-DMC and ct-DMC and their coexistence at equilibrium. The thermodynamic barrier (4.096 kcal/mol) determines the ratio of Li^+ -cc-DMC to Li^+ -ct-DMC. The thermodynamic barrier increases from 4.096 kcal/mol to 4.194 kcal/mol as the temperature increases from 298 K to 363 K (Supplementary Fig. 3), which is favorable to increase the proportion of ct-DMC. Therefore, increasing the temperature or increasing the heating time will change the ratio of Li^+ -cc-DMC to Li^+ -ct-DMC.”

Line 17-21, Page 31 “The Gibbs free energies can be calculated by the following equation: $G(T)=H(T)-TS(T)$

$$= \epsilon_{\text{ele}} + \text{ZPE} + G_{0 \rightarrow T}$$

Where G is Gibbs free energy, H is enthalpy, T is temperature, S is entropy, ϵ_{ele} is electronic energy, ZPE is acronym of zero-point energy. The analysis was conducted using Shermo software.⁶¹”

References

61. Lu, T., Chen, Q. Shermo: A general code for calculating molecular thermodynamic properties, *Comput. Theor. Chem.* **1200**, 113249 (2021).

Fig. 2a. Calculated free energy diagram for Li^+ -cc-DMC, Li^+ -ct-DMC and corresponding transition state at 298 K.

Supplementary Fig. 3. Calculated free energy diagram for Li⁺-cc-DMC, Li⁺-ct-DMC and corresponding transition state at 363 K.

Comments 2: The MD results are not reliable. The box is too small to get enough sampling, which can also be verified by the RDF plots. The $g(r)$ should be converged to 1 when r is large enough. A mistake may be made while analyzing the MD results. Besides, how are CC- and ct-DMC structures controlled in simulations? The MD results should deliver equilibrium sampling structures. The sharp peak of Li–O profiles is around 1.85 Å, which is even smaller than that in Li₂O crystals. Therefore, wrong or unsuitable simulation parameters may be adopted.

Response: Many thanks for your comments. We conducted ab initio molecular dynamics (AIMD) simulations instead of classical molecular dynamics (MD) simulations. For the classical MD simulation, large boxes are often used and contain thousands or more molecules to ensure sufficient sampling. However, for AIMD simulation, it obtains the electronic structure properties of the system at each given moment by solving the single-particle Schrödinger equation based on the Kohn-Sham orbit, which increases the amount of calculation sharply. Therefore, most electrolyte-related studies use boxes that are much smaller than that used in MD simulations, and the number of molecules introduced is also much smaller than that used in MD simulations. Recent AIMD studies on electrolyte properties included even fewer molecules than we used, but this was sufficient to ensure the accuracy and reliability of the simulation results (*Nature* **2022**, 608, 704-711; *Nat. Energy* **2018**, 3, 783-791; *Nat. Energy* **2022**, 7, 718-725).

Regarding the RDF plots, the reviewer commented that the $g(r)$ should be converged to 1 when r is large enough. When performing simulations with NPT ensemble, it is indeed the case since the volume of the box is fixed. However, we conducted AIMD simulations with NPT ensemble instead of the NVT ensemble. For the NPT system, $g(r)$ does not converge to 1 when r is large enough as the volume of the box will change during the simulation (*J. Phys. Chem. A* **2019**, 123, 8, 1689-1699; *Chem. Eng. Sci.* **2021**, 244, 116791; *Polymer* **2018**, 155, 136-145). We confirmed that our simulation results were correct after further checking the parameters used in our

simulations. Besides, we also conducted AIMD simulations using NVT ensemble as shown in Supplementary Fig. 10. RDFs of Li-O in cc-DMC and ct-DMC, FSI-O in cc-DMC and ct-DMC, and Li-O in cc-DMC and ct-DMC are converged to 1 when r is large enough. This also further verifies the reliability of the simulation parameters we used.

For MD simulations, the molecular structure can be set to "Flexible" or "Fixed". However, it does not need to be set for AIMD simulations since the force field in AIMD is generated by DFT calculation results, which is completely different from the force fields such as OPLS used in MD simulations. Besides, the AIMD can simulate the generation and cleavage of chemical bonds, but this cannot be achieved by MD simulations. Therefore, we performed AIMD instead of MD simulations.

In order to verify whether the Li-O bond length we calculated is correct, we conducted literature research and AIMD verification. The Li-O bond length in the Li_2O molecule is about 1.62 Å, which was confirmed by both experimental and theoretical calculations (*J. Chem. Phys.* **1963**, 39, 2463-2473; *Int. J. Quantum Chem.* **2013**, 113, 1264-1271; *Phys. Rev. B* **1996**, 53, 4989; *Molecular Physics.* **2009**, 107, 739-748). Compared to Li_2O molecules, the Li-O bond length in Li_2O crystals is slightly increased. For cubic Li_2O , the Li-O bond length can reach 2.02 Å (<https://next-gen.materialsproject.org/materials/mp-1960?formula=Li2O>). For lithium-ion battery electrolyte systems, the Li-O bond length is relevant with the electrolyte system. For example, the calculated Li-O bond length of LiFSI-butyl methyl ether (BME) electrolyte is about 1.85 Å (*Angew. Chem. Int. Ed.* **2023**, 62, e202310761), which is the same as our calculation results. For electrolytes such as LiFSI-dipropyl ether (DPE), LiFSI-diglyme (DIG) and Li-FSI-diethyl ether (DEE), the calculated Li-O bond length is about 1.8 Å (*Nat. Commun.* **2023**, 14, 868). Besides, the simulation results of the newly developed Lithium difluoro(oxalato)borate (LiDFOB)-cyclic ether (3,3,4,4,5,5-hexafluorotetrahydropyran, HFTHP), LiDFOB-bis(2,2,2-trifluoroethyl) ether (BTFE) and LiDFOB-1,1,2,2-tetrafluoroethyl-2,2,3,3-tetrafluoropropyl ether (TTE) electrolytes by Guo-Xing Li et al. showed that the Li-O bond length is also about 1.8 Å (*Nat. Energy* **2024**, 1-11. <https://doi.org/10.1038/s41560-024-01519-5>). Therefore, the length of Li-O bond in electrolyte is not necessarily longer than that in Li_2O crystal. Our newly added AIMD simulation using NVT ensemble indicates that the sharp peak of Li-O in DMC is still located at 1.85 Å (Supplementary Fig. 10g), which is the same as our previous AIMD simulation using NPT ensemble. In addition, we also repeatedly checked the calculation parameters to confirm the reliability of our calculation results. We thank the reviewers for their efforts in improving the reliability of our theoretical simulations, and we have added related discussion, the parameters and details of AIMD simulations in the revised manuscript as follows:

Line 18-23, Page 16; Line 1-2, Page 17 “Then, AIMD simulations with NPT ensemble are conducted to further verify the reliability of the simulation with NPT ensemble. As shown in Supplementary Fig. 10, both cases present similar coordination structures, and transitions between the two configurational isomers are also observed, which are consistent with previous simulations with NPT ensemble. When the two configuration isomers are coordinated with Li^+ , their primary solvating shell is still located at 1.85 Å. Similar to NPT system, the Li^+ -O in ct-DMC has higher intensity than the Li^+ -O in cc-

DMC, and the FSI-O in ct-DMC also shows much higher intensity than FSI-O in cc-DMC in NVT system.”

Line 11-14, Page 32 “A 10 ps simulation with NVT ensemble was performed after the NPT simulation. At temperatures of 298 K and 363 K, 50 ps in the NPT ensemble was performed to approach the equilibrium state, followed by 10 ps of NVT simulations to perform RDF analysis.”

Supplementary Fig. 10. AIMD simulation results using NVT ensemble. a, A snapshot of the simulation of 17 M LiFSI cc-DMC electrolytes. b, The representative solvation structure obtained from AIMD simulation. c, The specific solvation structures contain cc-DMC and ct-DMC in Fig. a and corresponding solvent configurations. d, A snapshot of the simulation of 17 M LiFSI ct-DMC electrolytes. e, The representative solvation structure obtained from AIMD simulation. f, The specific solvation structures contain cc-DMC and ct-DMC in Fig. d and corresponding solvent configurations. g-i, RDFs of Li-O in cc-DMC and ct-DMC (g), FSI-O in cc-DMC and ct-DMC (h), and Li-O in cc-DMC and ct-DMC (i).

Comment 3: If DMC structures are not fixed during MD structures, it is suggested to analyze their percentages under different temperatures and salt concentrations.

Response: Thanks so much for your valuable comments and suggestions. We conducted AIMD simulations on LiFSI-DMC electrolytes with different temperatures and salt concentrations according to your kind suggestions. Here, the initial solvents were all set to cc-DMC to analyze the effects of temperature and concentration on the transformation of cc-DMC to ct-DMC. Supplementary Fig. 11 shows the snapshot of the simulation results. At molar ratio of 1:15 (4 LiFSI and 60 cc-DMC are contained)

and temperature of 298 K, 2.7% of cc-DMC was converted to ct-DMC. As the molar ratio was increased to 2:15 (8 LiFSI and 60 cc-DMC are contained), the conversion ratio of cc-DMC to ct-DMC was increased to 3.5%. We further raised the temperature to 363 K to analyze its effect on the configurational isomerization transition while maintaining the same concentration. The conversion ratio of cc-DMC to ct-DMC was further increased to 3.8%. It should be noted that our simulation results did not reach the actual equilibrium state due to the limitations of computing resources and time. However, the change of configurational isomerization with temperature and concentration can still be well reflected. To better understand the effect of temperature on the transition from cc-DMC to ct-DMC, we calculated the Gibbs free energy of the molecule at 298 K and 363 K. As shown in Fig. 2a and Supplementary Fig. 3, kinetic barrier for the transformation of Li⁺-cc-DMC to Li⁺-ct-DMC decreases from 8.077 kcal/mol to 7.978 kcal/mol as the temperature increases from 298 K to 363 K. The thermodynamic barrier increases from 4.096 kcal/mol to 4.194 kcal/mol as the temperature increases from 298 K to 363 K, which is also favorable to increase the proportion of ct-DMC. Those thermodynamic calculation results also indicate that increasing the temperature would promote the transformation of cc-DMC to ct-DMC. We sincerely thank the reviewers for their valuable suggestions, which enable us to have a more in-depth analysis and discussion of the electrolyte studied. We have added related discussion in the revised manuscript as follows:

Line 2-8, Page 17 “Subsequently, we further investigated the effects of concentration as well as temperature on the configurational isomerization transitions using AIMD. At molar ratio of 1:15 (LiFSI:DMC) and temperature of 298 K, 2.7% of cc-DMC was converted to ct-DMC (See Supplementary Fig. 11). The conversion ratio of cc-DMC to ct-DMC was increased to 3.5% as the molar ratio was increased to 2:15. The conversion ratio can see a further increase (3.8%) as the temperature rises to 363 K, which is consistent with our spectral and DFT calculation results.”

Line 20-23, Page 7; Line 1-12, Page 8 “However, the cis-trans conformer becomes the most stable structure when taking the effect of Li⁺ into consideration for the real electrolytes (Fig. 2a). The Gibbs free energy of Li⁺-ct-DMC is 4.096 kcal/mol lower than that of Li⁺-cc-DMC. However, the transition from Li⁺-cc-DMC to Li⁺-ct-DMC requires crossing a high energy transition state. It is well known that an edge-CH₃ in cc-DMC can be transformed into ct-DMC by rotating 180° along the adjacent C-O bond. The energy reaches its maximum value as the -CH₃ rotates about 90° along the C-O bond, which is the transition state shown in Fig. 2a. In this case, the kinetic energy barrier for the transformation of Li⁺-cc-DMC to Li⁺-ct-DMC is 8.077 kcal/mol, and the kinetic energy barrier for the transformation of Li⁺-ct-DMC to Li⁺-cc-DMC is 12.174 kcal/mol. Actually, kinetic energy difference to cross the transition state leads to the difference in transformation kinetics between cc-DMC and ct-DMC and their coexistence at equilibrium. The thermodynamic barrier (4.096 kcal/mol) determines the ratio of Li⁺-cc-DMC to Li⁺-ct-DMC. The thermodynamic barrier increases from 4.096 kcal/mol to 4.194 kcal/mol as the temperature increases from 298 K to 363 K (Supplementary Fig. 3), which is favorable to increase the proportion of ct-DMC. Therefore, increasing the temperature or increasing the heating time will change the ratio of Li⁺-cc-DMC to Li⁺-

ct-DMC."

Supplementary Fig. 11. AIMD simulation results for LiFSI-DMC electrolytes with different concentrations and temperatures. a, A snapshot of the simulation of LiFSI-DMC with molar ratio of 1:15 at 298 K. 17 M LiFSI cc-DMC electrolytes. b, A snapshot of the simulation of LiFSI-DMC with molar ratio of 2:15 at 298 K. c, A snapshot of the simulation of LiFSI-DMC with molar ratio of 2:15 at 363 K. 17 M LiFSI cc-DMC electrolytes.

Fig. 2a. Calculated free energy diagram for Li⁺-cc-DMC, Li⁺-ct-DMC and corresponding transition state at 298 K.

Supplementary Fig. 3. Calculated free energy diagram for Li⁺-cc-DMC, Li⁺-ct-DMC and corresponding transition state at 363 K.

Comment 4: Therefore, the current simulation results can hardly support the conclusions of solvation structure analyses in this work. Besides, there are some other issues that should be further considered.

Figure 1a compares the solvent-to-salt ratio of various electrolytes. However, it should be noted that the calculation method of salt concentration can be different in different papers. Especially, the mix volume effect can have a significant influence on the calculated salt concentration.

Response: Thanks so much for your helpful comments. The reviewer's comments are very important in accurately describing the concentration of the electrolyte. Mix volume effect would greatly affect the volume of the electrolyte, especially for the highly concentrated electrolytes. Calculating electrolyte concentration based on solvent volume or based on the total volume of prepared electrolytes would lead to large differences. Therefore, it is necessary to give an accurate calculation method for the concentration of prepared electrolytes. The concentrations given in our article are based on the volume of solvent rather than the total volume of the electrolyte, which is generally used in previous studies (*Science*, **2015**, 350, 938-943; *Proc. Natl. Acad. Sci. U.S.A.* **2018**, 115, 1156-1161; *Chem*, **2018**, 4, 174-185). We have clarified the calculation method of concentration in the revised manuscript as follows:

Line 8-9, Page 7 "Note that, the electrolyte concentration is calculated based on the volume of solvent rather than the total volume of the electrolyte."

Comment 5: "A solvent-to-salt ratio of 1 seems to be the solubility limit of electrolytes." A solvent-to-salt ratio of 1 is highlighted. However, why such a ratio should be the limitation of solubility? The underlying chemistry and materials science should be explained.

Response: Thank you very much for your comments and suggestions. The conclusion that a solvent-to-salt ratio of 1 seems to be the solubility limit of electrolytes was obtained based on previous reports. We summarize the solvent-to-salt ratios that can

be achieved with highly concentrated electrolytes reported previously as shown in Fig. 1a. As far as we know, neither aqueous nor organic electrolytes have a solvent-to-salt ratio less than 1. Therefore, we conclude that the solvent-to-salt ratio of 1 seems to be the solubility limit of electrolytes. Such a description may not be rigorous enough, which may also lead to misunderstanding by readers. We have deleted the relevant description in the revised manuscript.

Comment 6: It is suggested to analyze the HOMO of different cc- and ct-DMC structures to demonstrate their electronic differences.

Response: Thanks so much for your helpful suggestions. The highest occupied molecular orbital (HOMO)-the lowest unoccupied molecular orbital (LUMO) structures of the cc-DMC and ct-DMC are shown in Supplementary Fig. 4. The calculated energy value of the HOMO level for the cc-DMC is -7.747 eV, which is much smaller than the ct-DMC (-7.704 eV). Therefore, the electrons at the HOMO level of ct-DMC molecules have higher energy, which makes it easier to combine with positively charged Li^+ and thus promote the improvement of electrolyte solubility. We have added related discussion in the revised manuscript as follows:

Line 13-15, Page 8 "The HOMO was also calculated for both cc-DMC and ct-DMC (Supplementary Fig. 4). The HOMO of the ct-DMC presents much higher energy than the cc-DMC, which enables ct-DMC more easily coordinated with positively charged Li^+ , thereby increasing the solubility of electrolytes."

Supplementary Fig. 4. HOMO-LUMO structures of the cc-DMC (a) and ct-DMC (b).

We sincerely hope our explanation can satisfy the reviewer on this question and are also sincerely looking forward to getting support from the reviewer.

Manuscript: NCOMMS-24-07587B

Title: “Conformational isomerism breaks the electrolyte solubility limit and stabilizes 4.9 V Ni-rich layered cathodes”

We thank the editor and reviewers for carefully assessing our manuscript (NCOMMS-24-07587B) and providing us the opportunity to further revise the manuscript in Nature Communications. We especially appreciate the insightful questions raised by the reviewers and welcome the opportunity to address these questions. The responses are listed point-by-point in the following contents, and revisions have been highlighted by **yellow color** in the revised manuscript. Following are our responses and detailed explanation towards these comments from the reviewers.

Responses to Reviewers:

To Reviewer #3:2-3
To Reviewer #4:4-14
To Reviewer #5:15

Response to Reviewers' comments

Reviewer #3:

General comments: Although the authors added a lot of calculations to revise the manuscript, previous issues have been well addressed. Therefore, I still do not think the current manuscript can meet the high standards of Nature Communications.

Response: We appreciate your efforts in evaluating the manuscript. We are very grateful for your constructive comments and suggestions on the AIMD simulation results. However, the focus of our research is experimental in nature. We did not include any AIMD results in our original manuscript. We initially tried to use AIMD simulation to briefly interpret the experimental results during revision. Obviously, this was not successful. The editorial team also recognizes that the main focus and achievement of this manuscript is experimental in nature. To avoid immature AIMD simulation results from reducing the quality of this study, the editorial team suggested that the AIMD simulation portion can be removed from the manuscript. Only minor changes have been made to the non-simulation parts, and the relevant changes have been highlighted. The detailed responses to the comments are shown below.

Comments 1: In previous Comments 1, "the thermodynamic barrier increases from 4.096 kcal/mol to 4.194 kcal/mol as the temperature increases from 298 K to 363 K". The thermostability energy difference between cc-DMC and ct-DMC is too small, which can not explain the obvious solubility salt differences among different temperatures.

Response: Thanks so much for your valuable comments. Interpreting the solvation structure of electrolytes is a major topic in battery research. The solvent structure in reality is extremely complex. The model we have currently constructed is only an ideal one. Therefore, it can only qualitatively analyze the effect of temperatures and configurational isomerism on solubility.

Comments 2: The NPT and NVT ensembles should deliver the same result when the structures are sampled enough. According to comments 2, the unreasonable RDF results may indicate that the structures were not sampled sufficient or there is a mistake during calculation or data analysis, which is not acceptable for a high-quality Nature Communications paper.

Response: Thanks so much for your valuable comments. As you said, only when the structures are sampled enough, the NPT and NVT ensembles would deliver the same result. Obviously, it does not apply to our simulation system. Of course, insufficient samples or mistake during calculation or data analysis may also exist in our simulations. In order to avoid any mistakes that may lead to misleading, we have removed AIMD simulation portion from the manuscript.

Comment 3: The review did not highlight the difference between classical MD and AIMD. The key point is that the results should be reasonable. It would be appreciated that the simulation results agree with the experimental results or the simulation results can explain the experiments well.

Response: Thanks so much for the helpful comments. Classical MD and AIMD have their own areas of application and expertise in solving problems. Using appropriate simulation methods, building correct models and reasonable analysis are also

necessary to obtain correct results. In many cases, the parameters and force fields considered in the simulation are not completely consistent, which leads to deviations from the experimental results. Although our simulation results can explain the experimental phenomena to a certain extent, the system we studied is a high-viscosity electrolyte, and we cannot be sure whether the established model and the obtained results are reasonable. In order to avoid misleading potential readers due to the slight inconsistency between experiments and simulations, we have deleted all simulation results.

Comment 4: In comment 3, the ratio of ct-DMC ranges from 3.5% to 3.8% from 298 K to 363 K. The ratio difference is very small, which can not explain the experimental results of conformational isomerism regulation.

Response: Thanks so much for the helpful comments. The models and analyses we built in our simulations are based on previous reports, which is usually an ideal system. However, the real electrolyte system is extremely complex. No simulation has yet been able to fully reflect the real electrolyte structure. In addition, the electrolyte system we studied not only involves configurational isomerism but also is a high viscosity and high aggregation system, which may cause it to deviate from the actual situation. Therefore, our simulation results can only qualitatively explain the experimental phenomenon. Of course, this qualitative result may be unacceptable to some potential research readers, especially those engaged in simulation work. Besides, the focus and highlight of our research lies in the experimental part. Therefore, all AIMD simulation results are removed from the manuscript.

Reviewer #4:

General comments: This manuscript presents a novel approach to enhancing the performance of lithium metal batteries by manipulating the solvent's configurational isomerism to create a highly concentrated electrolyte. By switching from a cis-cis to a cis-trans configuration of the DMC solvent, the authors successfully developed a 17 M electrolyte with a low solvent-to-salt ratio, resulting in a fully anion-mediated solvation structure with improved oxidative stability. This electrolyte significantly enhances the stability and performance of high-voltage Li||NCM811 batteries, even under extreme conditions, such as higher operational temperatures and high cut-off voltages.

However, the description of the simulations are not up to the standards of Nature Communications, so I cannot recommend the publication of this manuscript in this journal.

Response: Thank you very much for your comments and suggestions for further improving our manuscript. We are very grateful for your constructive comments on the MD simulation results. However, the focus of our research is experimental in nature. We did not include any MD results in our original manuscript. We initially tried to use MD simulation to briefly interpret the experimental results during revision. However, it was not successful due to lack of experience in simulation. The editorial team also recognizes that the main focus and achievement of this manuscript is experimental in nature. Following the editorial team's advice, we removed MD simulation parts from the manuscript. We carefully revised the manuscript for non-simulation parts, and the relevant changes have been highlighted. The detailed responses to the comments are shown below.

Major Issues

Comments 1: The reliability of the NVT ensemble is not described. In Fig. 4, which shows a "snapshot", there is considerable vacuum in the space. The authors simply say that "pressure was maintained under 1 atm during the simulation." More precise language is needed here. I assume they mean that the pressure was maintained at 1 atm, but it can also be interpreted that it was below this pressure. The authors describe that there were volume fluctuations during the trajectories, but there is no description of how large these fluctuations were. How closely did the density of the electrolyte match experimental values?

Response: Thanks so much for your valuable comments and questions. We apologize for not providing many simulation details. We conducted both NVT and NPT simulations. Fig. 4 shows the NPT simulation results. During the simulation, the pressure was maintained at 1 atm rather than below this pressure for the NPT ensemble. Since our simulation only includes 28 DMC and 40 LiFSI, the small sample and irregular shape prevent us from getting an accurate volume fluctuation range. Similarly, the density of the electrolyte can be easily obtained for NVT simulation, but it is difficult to get an accurate value for NPT.

The main focus and achievement of this manuscript is experimental in nature, and our original manuscript did not contain any simulation results. During the revision process we attempted to use AIMD simulations to explain some of the experimental results. However, we found that it is difficult to obtain reasonable results in the absence of

sufficient experience and for the system we studied that contains both structural isomers and high viscosity. In order to avoid presenting immature results and thus misleading readers, we have deleted the simulation part.

Comments 2: How was the cc-ct ratio monitored in the AIMD simulation? The short simulation time may have resulted in only a minor switch from cc to ct molecules in the trajectory, which may need to be acknowledged.

Response: Many thanks for your comments. We determined the cc-ct ratio by manual counting, since our simulation only included 28 DMC molecules and cc-DMC and ct-DMC can be well distinguished due to their large structural differences. We have to admit that this is not professional enough. From your professional comments and questions, we realize that theoretical simulation is a research direction that requires high professionalism. There are many parameters that we have not fully considered. As you said, our short simulation time may result in only a small amount of cc-DMC being transformed into ct-DMC. In order to avoid presenting immature results and emphasize the importance of the experimental part, we have deleted all simulation results.

Comment 3: Aggregation in the electrolyte is not visually distinguishable in Figure 4. (as mentioned in Fig 4a vs. 4d); the same is true for NVT (Fig. S10), which shows no agglomeration. It needs to be explained to be convincing. Additionally, could this have been a result of the NPT ensemble?

Response: Thanks so much for your valuable comments and questions. As shown in R1a, there are a lot of empty spaces inside the electrolyte (marked by a red oval). We think this may be caused by the aggregation of salts. However, for ct-DMC-based electrolytes, there is basically no such large empty space inside. The same situation can also be observed in the NVT simulation results (Figure R2). At the same time, we have to admit that this difference is not easy to detect. Moreover, this may be influenced by subjective factors and observation angles. Therefore, we deleted the simulation parts.

Fig. R1. NPT simulation results. a, A snapshot of the simulation of 17 M LiFSI cc-DMC electrolytes. b, The representative solvation structure obtained from AIMD simulation.

Fig. R2. NVT simulation results. a, A snapshot of the simulation of 17 M LiFSI cc-DMC electrolytes. b, The representative solvation structure obtained from AIMD simulation.

Comment 4: Even for an NPT-based simulation, RDF should approach 1. The authors' reason that this may not be the case for NPT is only valid if the distances are fixed in the starting image of the trajectory and, consequently, there is a significant shift in the volume during the equilibration process. For a given snapshot, it should go to 1. This may need to be acknowledged. Also, it should be mentioned if the RDF curves have been smoothed. The peaks are expected to be more spiky unless smoothed for a cell of this size.

Response: We are very grateful for the reviewers' professional comments. We would like to thank the reviewers again for popularizing the knowledge related to AIMD simulation so that we can be more professional and meticulous in future research. The RDF curves were not smoothed. We have to admit that our current AIMD simulations are not mature enough, although they can explain the experimental results to a certain extent. Here we would like to emphasize again that the focus and highlight of our research is the experimental part. To avoid immature AIMD simulation results from reducing the quality of this study, we have removed the simulation part.

Comment 5: Fig. S26 color/voltage legend needed?

Response: Thank you very much for your reminding. A clearer regulation has been added as shown below:

Supplementary Fig. 24. In-situ EIS during charging for 1 M LiPF₆-EC/DEC (a) electrolytes and ct/cc-0.82 electrolytes (b).

Comment 6: The equilibration criteria and related data for the trajectories need to be specified. Specifically, please clarify the phrase 'actual equilibrium state' in the statement "It should be noted that our simulation results did not reach the actual equilibrium state due to the limitations of computing resources and time." provided in the rebuttal for Review 2. This work will only be acceptable if the system is equilibrated computationally (energy and temperature) to extract conclusions from the calculated trajectory.

Response: Thanks so much for your helpful comments. After comprehensively reviewing your comments and suggestions, we believe it is more reasonable to delete the simulation part. First, simulations are not necessary to explain our experimental results since the original draft did not include any simulation results. In addition, we currently do not have sufficient computing resources to support our large simulations. Finally, the electrolyte system we studied involves a configurational isomerization transition and the electrolyte has an ultrahigh viscosity, which has never been reported before. We currently cannot handle this system very well from a simulation perspective. Therefore, in this study, we would like to present some of the peculiar chemical properties of the electrolyte only from an experimental perspective. Since this is a very interesting system, in subsequent research we may cooperate with experienced theoretical simulation researchers to analyze the properties of the electrolyte from a simulation perspective.

Minor Issues

Comment 7: Abstract - Line 20 - "Transitions between cis-cis and cis-trans conformers were visually observed through Nuclear Magnetic Resonance (NMR) testing": "Visually" from NMR did not make sense.

Response: Thanks so much for your helpful comments. We agree with you that using "Visually" doesn't make sense here. We have deleted it in the revised manuscript as

follows:

Line 20-21, Page 1 “Transitions between cis-cis and cis-trans conformers were observed through Nuclear Magnetic Resonance (NMR) testing.”

Comment 8: Fig. 1 has two panels labeled c)

Response: We are so sorry for our mistake. We have revised the Figure as shown below:

Fig. 1 | Comparison of electrolyte properties. **a**, Comparison of recently reported works about highly concentrated electrolytes. **b**, Images of various LiFSI-DMC electrolytes with different concentrations. **c**, Radar chart of the advantages of concentrated electrolytes constructed by manipulating cis-trans configuration (compared with ester and ether electrolytes). **d-e**, Failure mechanism of high voltage Li metal batteries using typical ester electrolytes (**d**) and high cis-trans electrolytes (**e**).

Comment 9: In the main text there is a description (on p. 8) saying the HOMO has a “much higher energy” in ct-DMC than in cc-DMC. I disagree that that 43 meV counts as a considerable difference. Also, regarding Supplementary Fig. 4, it shows side and top views of MOs, but it doesn’t say which orbitals we are looking at (HOMO or LUMO, or which is which).

Response: Thanks so much for your helpful comments. We also apologize for the imprecise wording. It is unreasonable to describe the energy difference of 43 eV as "much higher energy". We have deleted the "much" in the revised manuscript. In the Supplementary Fig. 4, what we focus on, and study is the HOMO orbital of the DMC molecule, since the HOMO orbital is related to the coordination ability of Li^+ . We apologize for not marking it clearly. We have clearly marked it in the revised Supplementary Fig. 4 as shown below:

Supplementary Fig. 4. Top view and side view HOMO structures of the cc-DMC (a) and ct-DMC (b).

Comment 10: The panel order in Fig. 3 is inconsistent (has a strange order).

Response: Thanks so much for your helpful comment. We have adjusted the panel order as shown below:

Fig. 3 | Identifying the evolution of conformers by NMR tests. **a**, ^1H spectra of electrolytes at different temperatures. The intensity of ^1H in D_2O is normalized. **b**, Schematic diagram explaining the meaning of ^1H NOESY NMR spectra. **c**, The fragment of ^1H - ^1H NOESY NMR spectrum of electrolytes at 333 K in D_2O . **d**, The fragment of ^1H - ^1H NOESY NMR spectrum of electrolytes at 363 K in D_2O . **e**, ^1H spectra of electrolytes before and after NOESY tests at 333 K. **f**, ^1H spectra of electrolytes before and after NOESY tests at 363 K.

Comment 11: In supplementary Fig. 10 caption there is a reference to “Fig. a” – I believe the number is missing.

Response: Thanks so much for your reminding. In supplementary Fig. 10 caption, the reference is referred to “Fig. 10a”. Figure c and the figure a I want to refer are both in Figure 10, so the numbers are omitted.

Comment 12: p9 152 - How this ct/cc ratio value was obtained needs to be clarified. Was it from the O-CH3 stretching of the DMC Raman spectra?

Response: Thanks so much for your helpful question. The ct/cc ratio was obtained

based on the O-CH₃ stretching fitting results of the DMC Raman spectra. We have clarified it in the revised manuscript as follows:

Line 7-9, Page 9 “The ratio of bonded cis-trans DMC/bonded cis-cis DMC can be obtained based on the O-CH₃ stretching fitting results, and the ratio increases with concentration as shown in the Fig. 2d.”

Comment 13: p32 578 - Energy cutoff for the AIMD - 500 Ry. Typo? (50 Ry?)

Response: We are so sorry for our typo. The cutoff energy for the AIMD is 50 Ry.

Comment 14: p7 101 - remove "a" and "conformer".

Response: Thanks so much for your reminding. We have revised the sentence as shown below:

Line 5-6, Page 7 “The configuration of DMC gradually transforms into a cis-trans conformer from cis-cis upon thermal triggering.”

Comment 15: p8 139 - increasing,

Response: Thanks so much for your reminding. We have revised the sentence as shown below:

Line 9-11, Page 8 “The thermodynamic barrier increases from 4.096 kcal/mol to 4.194 kcal/mol as the temperature increasing from 298 K to 363 K (Supplementary Fig. 3), which is favorable to increase the proportion of ct-DMC.”

Comment 16: p9 151 - cis-trans

Response: Thanks so much for your reminding. We have revised it in the revised manuscript as shown below:

Line 11-13, Page 9 “In order to present the relative proportion of cis-trans and cis-cis in the electrolyte, the 1 M, 10 M, and 17 M electrolytes are marked as ct/cc-0.10, ct/cc-0.59, and ct/cc-0.9, respectively.”

Comment 17: p11 Fig 2 - switch 0.59 with 0.35 pie charts for clarity or typo?

Response: Thanks so much for your reminding. We reversed the order of the pie charts for ct/cc-0.1 and ct/cc-0.9 so that they correspond to the results in Figure 2e.

Fig. 2 | Solvent structure analysis of ultra concentrated electrolytes constructed by conformers. a, Transition state energies of different conformers of DMC solvent in the presence of Li^+ . b, Schematic diagram of the interconversion of different conformers. c, The fitting results of Raman spectra (O- CH_3 stretching of the DMC) of various electrolytes. d, Relationship between the salt concentration (x-axis) and bonded ct-DMC/cc-DMC (y-axis). e, The fitting results of Raman spectra (S-N stretching of the FSI $^-$) of various electrolytes. The positions of the dashed lines and arrows in Fig. c and e correspond to peak positions of the relevant solvation structures. f, Pie chart of solvation structure composition of different electrolytes.

Comment 18: p16 280 - NVT?

Response: We are sorry for the typo. We verified the reliability of the simulation of NPT ensemble with NVT ensemble. We have revised it. Thanks again.

Comment 19: p5 75 - Minor - Fig c is referred to before b

Response: Thanks so much for your reminding. We have reversed the positions of Figure 1b and Figure 1c as shown below:

Fig. 1 | Comparison of electrolyte properties. **a**, Comparison of recently reported works about highly concentrated electrolytes. **b**, Images of various LiFSI-DMC electrolytes with different concentrations. **c**, Radar chart of the advantages of concentrated electrolytes constructed by manipulating cis-trans configuration (compared with ester and ether electrolytes). **d-e**, Failure mechanism of high voltage Li metal batteries using typical ester electrolytes (**d**) and high cis-trans electrolytes (**e**).

Comment 20: p19 320 Fig S14 - Cannot distinguish the curves. A legend is needed for the colors used for M LiPF₆ EC/DEC electrolytes and ct/cc 0.82 electrolytes.

Response: Thanks so much for your reminding. We have added legend in revised Supplementary Figure as shown below:

Supplementary Fig. 12. (a) Capacity comparison at different charging cut-off voltages using 1 M LiPF₆-EC/DEC electrolytes and ct/cc-0.82 electrolytes. (b) Charging curves under different charging cut-off voltages for the battery with ct/cc-0.82 electrolytes.

Reviewer #5:

General comments: I co-reviewed this manuscript with one of the reviewers who provided the listed reports. This is part of the Nature Communications initiative to facilitate training in peer review and to provide appropriate recognition for Early Career Researchers who co-review manuscripts.

Response: Thank you very much for co-reviewing this paper. Your active contribution is extremely important to improve the quality of this manuscript. Thanks again.